# TangoFlux: Super Fast and Faithful Text to Audio Generation with Flow Matching and Clap-Ranked Preference Optimization

**Chia-Yu Hung**[1]    **Navonil Majumder**[1]    **Zhifeng Kong**[2]    **Ambuj Mehrish**[3]
**Amir Ali Bagherzadeh**[4]    **Chuan Li**[4]    **Rafael Valle**[2]    **Bryan Catanzaro**[2]    **Soujanya Poria**[1]

[1]Nanyang Technology University (NTU) [2]NVIDIA
[3]Ca' Foscari University of Venice
[4]Lambda Labs

```
chiayu001@e.ntu.edu.sg
{navonil.majumder,soujanya.poria}@ntu.edu.sg
ambuj.mehrish@unive.it
{zkong, rafaelvalle, bcatanzaro}@nvidia.com
{amirali.zadeh, c}@lambdal.com
```

## Abstract

We introduce **TangoFlux**, an efficient Text-to-Audio (TTA) generative model with 515M parameters, capable of generating up to 30 seconds of 44.1kHz audio in 3.7 seconds on a A40 GPU. A key challenge in aligning TTA models lies in creating preference pairs, as TTA lacks structured mechanisms like verifiable rewards or gold-standard answers available for Large Language Models (LLMs). To address this, we propose CLAP-Ranked Preference Optimization (CRPO), a novel framework that iteratively generates and optimizes preference data to enhance TTA alignment. We show that the audio preference dataset generated using CRPO outperforms the static alternatives. With this framework, **TangoFlux** achieves state-of-the-art performance across both objective and subjective benchmarks. `https://tangoflux.github.io/` holds the model-generated audio samples for comparison.

## 1 Introduction

Audio is integral to daily life and creative industries, from enhancing communication and storytelling to enriching experiences in music, sound effects, and podcasts. However, creating high-quality audio, such as foley effects or music compositions, demands significant effort, expertise, and time. Recent advancements in text-to-audio (TTA) generation (Majumder et al., 2024; Ghosal et al., 2023; Liu et al., 2023; 2024b; Xue et al., 2024; Vyas et al., 2023; Huang et al., 2023b;a) enabled automatic and rapid creation of diverse and expressive audio content directly from textual descriptions. This technology holds immense potential to accelerate audio production and creative multimedia workflows. However, many existing models face challenges with controllability, often struggling to fully capture the details in the input prompts, especially when the prompts are complex, containing many events with diverse temporal relationships. This sometimes results in audios that omit certain events or place the events in a wrong order. At times, the generated audio may even contain input-adjacent, but unmentioned and unintended, events, that could be characterized as hallucinations.

Alignment often leverages reinforcement learning from human feedback (RLHF) or other reward-based optimization methods to endow the generated outputs with human preferences, ethical considerations, and task-specific requirements (Ouyang et al., 2022). Recently Majumder et al. (2024) employed alignment for TTA model training. One critical challenge in implementing alignment for TTA lies in the creation of preference pairs. Unlike LLM alignment, where off-the-shelf reward models (Lambert et al., 2024a;b) and human feedback data or verifiable gold answers are available, TTA domain as yet lacks such tooling.

While audio language models (Chu et al., 2024; 2023; Tang et al., 2024) can take audio inputs and generate textual outputs, they usually produce noisy feedback, unfit for preference pair creation for audio. BATON (Liao et al., 2024) employs human annotators to assign a binary label 0/1 to each audio sample based on its alignment with a given prompt. However, such labor-intensive manual approach is often impractical at a large scale.

To address these issues, we propose CLAP-Ranked Preference Optimization (CRPO), a simple yet effective approach to generate audio preference data and perform preference optimization on rectified flows. As shown in Fig. 1, CRPO consists of iterative cycles of audio sampling, preference pair curation, and preference optimization, resembling a self-improvement algorithm. A notable aspect of this approach is evolution by generating its own training dataset, dynamically aligning itself over multiple iterations. We show the CLAP (Wu* et al., 2023) can serve as a proxy reward model for ranking generated audios by alignment with the text description. With this ranking, we construct an audio preference dataset that post alignment yields superior performance to other static audio preference datasets, such as, BATON and Audio-Alpaca (Majumder et al., 2024).

Unlike many closed TTA models (Evans et al., 2024b;a; Copet et al., 2024) that are trained on proprietary data, our open-source TANGOFLUX is trained on open data, achieving *state-of-the-art* performance on benchmarks and out-of-distribution human evaluation, despite its smaller size. TANGOFLUX also supports variable-duration audio generation up to 30 seconds with an inference time of 3.7 seconds on an A40 GPU. This is unlike diffusion-based TTA models (Ghosal et al., 2023; Majumder et al., 2024; Liu et al., 2024b) that are known to require too many denoising steps for a decent output, consuming much compute and time. This is achieved using a transformer (Vaswani et al., 2023) backbone that undergoes pretraining, fine-tuning, and preference optimization with rectified flow matching training objective.

**Our contributions**:

(i) We introduce TANGOFLUX, a small TTA model based on rectified flow with *state-of-the-art* performance for fully non-proprietary training data.
(ii) We propose CRPO, a simple yet effective strategy for dynamically generating audio preference data and aligning rectified flows. By iteratively refining the preference data, CRPO continuously improves itself, outperforming static audio preference datasets.
(iii) We conduct extensive experiments and highlight the importance of each component of CRPO in aligning rectified flows for improving scores on benchmarks.
(iv) We will release the code and model weights.

## 2 METHOD

TANGOFLUX consists of FluxTransformer blocks, which are composed of Diffusion Transformer (DiT) (Peebles & Xie, 2023) and Multimodal Diffusion Transformer (MMDiT) (Esser et al., 2024) conditioned on a text prompt and a duration embedding to generate 44.1kHz audios of up to 30 second long. TANGOFLUX learns a rectified flow trajectory to the latent audio representation encoded by a variational autoencoder (VAE) (Kingma & Welling, 2022). As shown in Fig. 1, the training pipeline consists of two stages: pre-training and alignment. TANGOFLUX is aligned with our CRPO method which iteratively generates new synthetic data and constructs preference pairs for preference optimization.

### 2.1 AUDIO ENCODING

We use the VAE from Stable Audio Open (Evans et al., 2024c), which is capable of encoding 44.1kHz stereo audio waveforms into latent representations. Given a stereo audio $X \in \mathbb{R}^{2 \times d \times sr}$ with $d$ as the duration and $sr$ as the sampling rate, the VAE encodes $X$ into a latent representation $Z \in \mathbb{R}^{L \times C}$, with $L$ and $C$ being the latent sequence length and channel size, respectively. The VAE decodes the latent representation $Z$ into the original stereo audio $X$. The entire VAE is kept frozen during TANGOFLUX training.

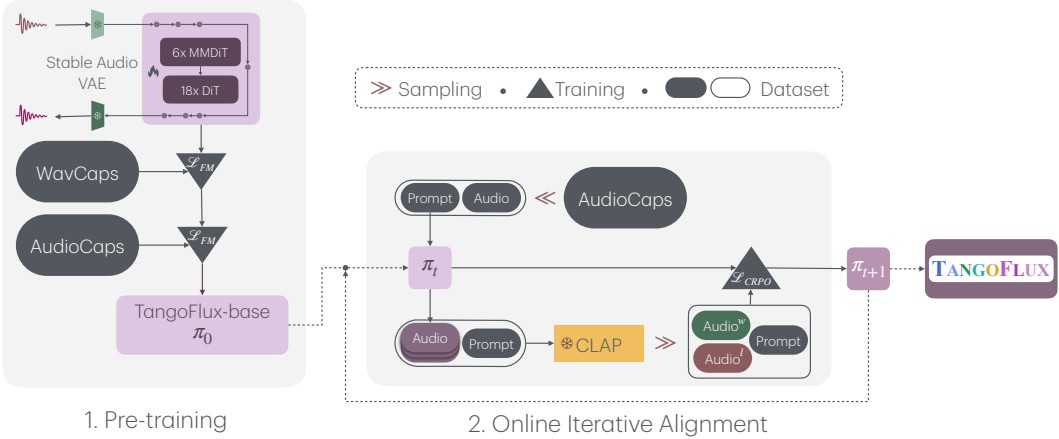

Figure 1: A depiction of the overall training pipeline of TANGOFLUX.

## 2.2 MODEL CONDITIONING

To control the generation of audio of varying lengths, we employ (i) text conditioning to control the content of the generated audio and (ii) duration conditioning to dictate the output audio length, up to a maximum of 30 seconds.

**Text Conditioning.** We obtain an encoding $c_{text}$ of the given textual description from a pretrained text-encoder. Given the strong performance of FLAN-T5 (Chung et al., 2022; Raffel et al., 2023) as conditioning in text-to-audio generation (Majumder et al., 2024; Ghosal et al., 2023), we select FLAN-T5 as our text encoder.

**Duration Encoding.** Inspired by the recent works (Evans et al., 2024c;a;b), to generate audios with variable length, we use a small neural network to encode the audio duration into a duration embedding $c_{dur}$ that is concatenated with the text encoding $c_{text}$ and fed into TANGOFLUX to control the duration of audio output. Crucially, TANGOFLUX always operates on a fixed-length latent space (30s of audio). The duration conditioning $c_{dur}$ explicitly controls how much of this fixed-length latent space contains actual audio content, and how much is dedicated to silence.

## 2.3 MODEL ARCHITECTURE

Following the recent success of FLUX models in image generation[1], we adopt a hybrid MMDiT and DiT architecture as the backbone for TANGOFLUX. While MMDiT blocks demonstrated a strong performance, simplifying some of them into single DiT block improved scalability and parameter efficiency[2]. These lead us to select a model architecture with 6 blocks of MMDiT, followed by 18 blocks of DiT. Each block has 8 attention heads of 128 width, totaling a width of 1024. This setting amounts to 515M trained parameters.

## 2.4 FLOW MATCHING

Several generative models have been successfully trained under the diffusion framework (Ho et al., 2020; Song et al., 2022; Liu et al., 2022). However, this approach is known to be sensitive to the choice of noise scheduler, which may significantly affect performance. In contrast, the flow matching (FM) framework (Lipman et al., 2023; Albergo & Vanden-Eijnden, 2023) has been shown to be more robust to the choice of noise scheduler, making it a preferred choice in many applications, including text-to-audio (TTA) and text-to-speech (TTS) tasks (Liu et al., 2024a; Le et al., 2023; Vyas et al., 2023).

Flow matching builds upon the continuous normalizing flows framework (Onken et al., 2021). It generates samples from a target distribution by learning a time-dependent vector field that maps

---

[1] https://blackforestlabs.ai/
[2] https://blog.fal.ai/auraflow/

samples from a simple prior distribution (e.g., Gaussian) to a complex target distribution. Prior work in TTA, such as AudioBox (Vyas et al., 2023) and Voicebox (Le et al., 2023), has predominantly adopted the Optimal Transport conditional path proposed by (Lipman et al., 2023). However, we utilize rectified flows (Liu et al., 2022) instead, which is a straight line path from noise to distribution, corresponding to the shortest path. See Appendix A.4 for the details on rectified flow.

**Inference.** For inference, we sample a noise $\tilde{x}_0 \sim \mathcal{N}(\mathbf{0}, \mathbf{I})$ and use Euler solver to compute $x_1$, based on the model-predicted velocity $u(\cdot; \theta)$ at each time step $t$. The results for the Heun solver are presented in Appendix A.7.

## 2.5 CLAP-RANKED PREFERENCE OPTIMIZATION

CLAP-Ranked Preference Optimization (CRPO) leverages a text-audio joint-embedding model like CLAP (Wu* et al., 2023) as a proxy reward model to rank the generated audios by similarity with the input description and subsequently construct the preference pairs.

Tango2 (Majumder et al., 2024) and CLIP-DPO (Ouali et al., 2024) curate preference pairs via prompt perturbation, sampling multiple outputs for each prompt to establish static rankings. However, these approaches rely on static preference datasets, which can limit alignment generalization and adaptability. In contrast, we adopt a dynamic preference optimization strategy: it generates new synthetic preference pairs at the start of each training iteration. This online data generation allows the model to continually refine its alignment through training which we show to further improve performance.

We set $\pi_0$ to a pre-trained checkpoint **TANGOFLUX**-base to align. Thereafter, CRPO iteratively aligns checkpoint $\pi_k \coloneqq u(\cdot; \theta_k)$ into checkpoint $\pi_{k+1}$, starting from $k = 0$. Each alignment iteration consists of three steps: (i) batched online data generation, (ii) reward estimation and preference dataset creation, and (iii) fine-tuning $\pi_k$ into $\pi_{k+1}$ via direct preference optimization. This alignment process allows the model to continuously self-improve by generating and leveraging its own preference data.

This approach of alignment is inspired by a few LLM alignment approaches (Zelikman et al., 2022; Kim et al., 2024a; Yuan et al., 2024; Pang et al., 2024). However, there are key distinctions to our work: (i) we align rectified flows for audio generation, rather than autoregressive language models; (ii) while LLM alignment benefits from numerous off-the-shelf reward models (Lambert et al., 2024b), which ease the construction of preference datasets based on reward scores, LLM judged outputs, or programmatically verifiable answers, the audio domain lacks such models or method for evaluating audio. We demonstrate that the CLAP model can serve as an effective proxy audio reward model, enabling the creation of preference datasets (see Appendix A.3). Finally, we highlight the necessity of generating online data at every iteration, as iterative optimization on offline data leads to quicker performance saturation and subsequent degradation.

### 2.5.1 CLAP AS A REWARD MODEL

CLAP reward score is calculated as the cosine similarity between textual and audio embeddings encoded by the model. Thus, we assume that CLAP can serve as a reasonable proxy reward model for evaluating audio outputs against the textual description. In Appendix A.3, we demonstrate that using CLAP as a judge to choose the best-of-N inferred policies improves performance in terms of objective metrics.

### 2.5.2 BATCHED ONLINE DATA GENERATION

To construct a preference dataset at iteration $k$, we first sample a set of prompts $M_k$ from a larger pool $B$. Then, we generate $N$ audios for each prompt $y_i \in M_k$ using $\pi_k$ and use CLAP[3] (Wu* et al., 2023) to rank those audios by similarity with $y_i$. For each prompt $y_i$, we select the highest-rewarded or -ranking audio $x_i^w$ as the winner and the lowest-rewarded audio $x_i^l$ as the loser, yielding a preference dataset $\mathcal{D}_k = \{(x_i^w, x_i^l, y_i) \mid y_i \in M_k\}$.

---

[3]`https://huggingface.co/lukewys/laion_clap/blob/main/630k-audioset-best.pt`

### 2.5.3 Preference Optimization

Direct preference optimization (DPO) (Rafailov et al., 2024c) is shown to be effective at instilling human preferences in LLMs (Ouyang et al., 2022). Consequently, DPO is successfully translated into DPO-Diffusion (Wallace et al., 2023) for alignment of diffusion models. The DPO-diffusion loss is defined as

$$
\begin{aligned}
L_{\text{DPO-Diff}} = -\mathbb{E}_{n,\epsilon^w,\epsilon^l} \log \sigma \Big( &- \beta \Big[ \|\epsilon_n^w - \epsilon_\theta(x_n^w)\|_2^2 - \|\epsilon_n^w - \epsilon_{\text{ref}}(x_n^w)\|_2^2 \\
&- \|\epsilon_n^l - \epsilon_\theta(x_n^l)\|_2^2 + \|\epsilon_n^l - \epsilon_{\text{ref}}(x_n^l)\|_2^2 \Big] \Big).
\end{aligned}
\tag{1}
$$

$n \sim U(0,T)$ is a diffusion step among $T$ steps; $x_n^l$ and $x_n^w$ represent the losing and winning audios, with $\epsilon \sim \mathcal{N}(0,\mathbf{I})$.

Following Esser et al. (2024), DPO-Diffusion loss is applicable to rectified flow through the equivalence (Lipman et al., 2023) between $\epsilon_\theta$ and $u(\cdot;\theta)$, thereby the noise matching loss terms can be substituted with flow matching terms:

$$
L_{\text{DPO-FM}} = -\mathbb{E}_{t\sim\mathcal{U}(0,1),x^w,x^l,y} \log \sigma \Big( - \beta \Big[ \underbrace{\|u(x_t^w,y,t;\theta) - v_t^w\|_2^2}_{\text{Winning loss}} - \underbrace{\|u(x_t^l,y,t;\theta) - v_t^l\|_2^2}_{\text{Losing loss}}
$$

$$
- \underbrace{\|u(x_t^w,y,t;\theta_r) - v_t^w\|_2^2}_{\text{Winning reference loss}} + \underbrace{\|u(x_t^l,y,t;\theta_r) - v_t^l\|_2^2}_{\text{Losing reference loss}} \Big] \Big), \tag{2}
$$

where $t$ is a flow matching timestep and $x_t^l$ and $x_t^w$ represent losing and winning audio, respectively.

The DPO loss for LLMs models the relative likelihood of the winner and loser responses, allowing minimization of the loss by increasing their margin, even if both log-likelihoods decrease (Pal et al., 2024). As DPO optimizes the relative likelihood of the winning responses over the losing ones, not their absolute values, convergence actually requires both likelihoods to decrease despite being counterintuitive (Rafailov et al., 2024b). The decrease in likelihood does not necessarily decrease performance, but required for improvement (Rafailov et al., 2024a). However, in the context of rectified flows, this behavior is less clear due to the challenges in estimating the likelihood of generating samples with classifier-free guidance (CFG). A closer look at $\mathcal{L}_{\text{DPO-FM}}$ (Eq. (2)) reveals that it can similarly be minimized by increasing the margin between the winning and losing losses, even if both losses increase. In Section 4.5, we demonstrate that preference optimization of rectified flows via $\mathcal{L}_{\text{DPO-FM}}$ suffer from this phenomenon as well.

To remedy this, we directly add the winning loss to the optimization objective to prevent *winning loss* from increasing:

$$
\mathcal{L}_{\text{CRPO}} := \mathcal{L}_{\text{DPO-FM}} + \mathcal{L}_{\text{FM}}, \tag{3}
$$

where $\mathcal{L}_{\text{FM}}$ is the flow matching loss computed on the winning audio as shown in Eq. (5). While the DPO loss is effective at improving preference rankings between chosen and rejected audio, relying on it alone can lead to overoptimization. This can distort the semantic and structural fidelity of the winning audio, causing the model's outputs to drift from the desired distribution. Adding the $\mathcal{L}_{\text{FM}}$ component mitigates this risk by anchoring the model to the high-quality attributes of the chosen data. This regularization stabilizes training and preserves the essential properties of the winning examples, ensuring a balanced and robust optimization process. Our empirical results demonstrates $\mathcal{L}_{\text{CRPO}}$ outperform $\mathcal{L}_{\text{DPO-FM}}$ as shown in Section 4.5.

## 3 Experiments

### 3.1 Model Training

We pretrained **TangoFlux** on Wavcaps (Mei et al., 2024) dataset for 80 epochs with the AdamW (Loshchilov & Hutter, 2019), $\beta_1 = 0.9, \beta_2 = 0.95$, a max learning rate of $5 \times 10^{-4}$. During the alignment phase, we used the same optimizer, but an overall batch size of $48$ and a maximum learning rate of $10^{-5}$. See Appendix A.5 for the complete training details.

## 3.2 DATASETS

**Training dataset**. We use complete open source data which consists of approximately 400k audios from *Wavcaps* (Mei et al., 2024) and 45k audios from the training set of *AudioCaps*. (Kim et al., 2019). Audios shorter than 30 seconds are padded with silence to 30s. Longer than 30 second audios are center cropped to 30 seconds. As the audio files are mono, we duplicated the channel to create pseudostereo audios for compatibility with the VAE.

**CRPO dataset.** We initialize the prompt bank as the prompts of *AudioCaps* training set, with a total of 45k prompts. At the start of each iteration of CRPO, we randomly sample 20k prompts from the prompt bank and generate 5 audios per prompt, and use the CLAP model to construct 20k preference pairs.

**Evaluation dataset.** For the main results, we evaluated TANGOFLUX on the *AudioCaps* test set, using the same 886-sample split as (Majumder et al., 2024). Objective metrics are reported on this subset. Additionally, we categorized *AudioCaps* prompts using GPT-4 to identify those with multiple distinct events, such as "Birds chirping and thunder strikes," which includes "sound of birds chirping" and "sound of thunder." Subjective evaluation was conducted on an out-of-distribution dataset with 50 challenging prompts.

## 3.3 OBJECTIVE EVALUATION

**Baselines.** We compare TANGOFLUX to few existing strong text-to-audio generation baselines: `Tango`, `Tango 2`, `AudioLDM 2`, `Stable Audio Open`, `AudioX`, and `GenAU-Full-L`, including the previous SOTA models. Across the baselines, we use the default recommended classifier-free guidance (CFG) scale (Ho & Salimans, 2022) and number of steps. For TANGOFLUX, we use a CFG scale of 4.5 and 50 steps for inference. Since TANGOFLUX and Stable Audio Open allow variable audio generation length, we set the duration conditioning to 10 seconds and use the first 10 seconds of generated audio to perform the evaluation. We also report the effect of CFG scale in the Appendix A.6.

**Evaluation metrics.** We evaluate TANGOFLUX using both objective and subjective metrics. Following (Evans et al., 2024a), we report the five objective metrics: Kernel Audio Distance (KAD)(Chung et al., 2025), Fréchet Distance ($FD_{open13}$) (Cramer et al., 2019), Kullback–Leibler divergence ($KL_{passt}$) , $CLAP_{score}$, and Inception Score (IS) (Salimans et al., 2016). These metrics allow high-quality audio evaluation up to 48kHz. Further details on these metrics are presented in Appendix A.10.

## 3.4 HUMAN EVALUATION

Following prior studies (Ghosal et al., 2023; Majumder et al., 2024), our subjective evaluation covers two key attributes of the generated audio: overall audio quality (OVL) and relevance to the text input (REL). OVL captures the general sound quality, including clarity and naturalness, ignoring the alignment with the input prompt. In contrast, REL quantifies the alignment of the generated audio with the provided text input. At least four annotators rate each audio sample on a scale from 0 (worst) to 100 (best) on both OVL and REL. This evaluation is performed on 50 GPT4o-generated and human-vetted prompts and reported in terms of three metrics: $z$-score, Ranking, and Elo score. The evaluation instructions, annotators, and metrics are in Appendix A.12. Due to resource constraint, we were not able to peform human evaluation on all the models.

## 4 RESULTS

### 4.1 MAIN RESULTS

Table 1 objectively compares TANGOFLUX with prior text-to-audio generation models on *AudioCaps*. Our results suggest that TANGOFLUX consistently outperforms the prior works on all objective metrics, except `Tango 2` on $KL_{passt}$ and $FD_P$. We hypothesize this specific discrepancy is attributable to the different sampling rates used for the different FD metrics. $FD_P$ operates at 16 kHz, hence it is required to downsample TANGOFLUX's generated audio, removing all of its high-frequency details. Conversely, the metric $FD_{open13}$ is evaluated directly at the 48 kHz sampling rate being sensitive to the

Table 1: Comparison of text-to-audio models. Output length represents the duration of the generated audio. Objective metrics include $FD_{openl3}$ for Fréchet Distance measured with OpenL3, $FD_P$ for for Fréchet Distance measured with PANNs, $KL_{passt}$ for KL divergence, and $CLAP_{score}$ for alignment.

| Model | #Params. | Duration | Steps | $FD_P \downarrow$ | $FD_{openl3} \downarrow$ | $KL_{passt} \downarrow$ | $KAD \downarrow$ | $CLAP_{score} \uparrow$ | $IS \uparrow$ | Inference Time (s) |
|---|---|---|---|---|---|---|---|---|---|---|
| **Few step sampling models** | | | | | | | | | | |
| ConsistencyTTA | 559M | 10 sec | 1 | 20.9 | 94.6 | 1.43 | 0.61 | 0.377 | 9.1 | <0.2 |
| AudioLCM | 160M | 10 sec | 1 | 19.2 | 107.4 | 1.58 | 0.56 | 0.363 | 10.2 | <0.2 |
| AudioLDM 2-large | 712M | 10 sec | 200 | 33.2 | 108.3 | 1.81 | 1.78 | 0.419 | 7.9 | 24.8 |
| Make-An-Audio 2 | **160M** | 10 sec | 100 | **15.6** | 98.7 | 1.33 | 0.45 | 0.406 | 9.4 | **2.3** |
| EzAudio-XL | 874M | 10 sec | 200 | 15.8 | 84.7 | 1.20 | **0.15** | 0.460 | 10.8 | 12.2 |
| Stable Audio Open | 1056M | 47 sec | 100 | 42.6 | 89.2 | 2.58 | 4.15 | 0.291 | 9.9 | 8.6 |
| Tango | 866M | 10 sec | 200 | 24.5 | 107.9 | 1.20 | 1.71 | 0.407 | 7.8 | 22.8 |
| Tango 2 | 866M | 10 sec | 200 | 20.8 | 108.4 | **1.11** | 1.38 | 0.447 | 9.0 | 22.8 |
| GenAU-Full-L | 1.25B | 10 sec | 100 | 20.1 | 93.2 | 1.37 | 0.96 | 0.447 | 12.0 | 5.3 |
| AudioX | 1.1B | 10 sec | 250 | 25.2 | 77.6 | 1.56 | 1.30 | 0.380 | 10.0 | 9.6 |
| TANGOFLUX-base | 516M | 30 sec | 50 | 20.7 | 80.2 | 1.22 | 0.67 | 0.431 | 11.7 | 3.7 |
| TANGOFLUX | 516M | 30 sec | 50 | 20.3 | **75.1** | 1.15 | 0.60 | **0.480** | 12.2 | 3.7 |

higher frequency details, where TANGOFLUX shows advantage for being able to directly generate audios at 44.1Khz.

## 4.2 HUMAN EVALUATION RESULTS

The results of the human evaluation are presented in Table 2, with detailed comparisons of the models across the evaluated metrics: z-scores, rankings, and Elo scores for both overall audio quality (OVL) and relevance to the text input (REL).

**z-scores:** z-score mitigates individual scoring biases by normalization into a standard normal variable with zero mean and one standard deviation. TANGOFLUX demonstrated the highest performance across both metrics, with z-scores of 0.2486 for OVL and 0.6919 for REL. This indicates its superior quality and strong alignment with the input prompts. Conversely, AudioLDM 2 scored the lowest with z-scores of -0.3020 (OVL) and -0.4936 (REL), suggesting both lower sound quality and weaker adherence to textual inputs as compared to the other models.

**Ranking:** Ranks provide an ordinal measure of performance, complementing z-score. TANGOFLUX achieved the best rank with a mean rank of 1.7 (OVL) and 1.1 (REL), and mode ranks of 2 (OVL) and 1 (REL), affirming its superiority in subjective evaluations. In contrast, AudioLDM 2 consistently ranked lowest, with mean ranks of 3.5 (OVL) and 3.7 (REL), and mode ranks of 4 for both metrics. StableAudio and Tango 2 had similar mean ranks for OVL (2.4), but Tango 2 outperformed StableAudio on REL (mean ranks: 1.9 vs 3.3). Notably, StableAudio's bimodal OVL ranks (modes 1 and 3) suggest polarized annotator perceptions, likely due to misalignment between prompts and outputs, as reflected in its REL rankings (mean 3.3, mode 3).

**Elo Scores:** Elo scores provide a probabilistic measure of model performance, by accounting for pairwise relative performance. Here, TANGOFLUX again excelled, achieving the highest Elo scores for both OVL (1,501) and REL (1,628). The Elo results highlight the robustness of TANGOFLUX, as it consistently outperformed other models in pairwise comparisons. Tango 2 emerged as the second-best performer, with Elo scores of 1,419 (OVL) and 1,507 (REL). StableAudio follows, showing competitive performance in OVL (1,444), but a weaker REL score (1,268). Like other metrics, AudioLDM 2 ranked last with the least Elo scores (1,236 for OVL and 1,196 for REL).

## 4.3 CRPO BEATS STATIC PREFERENCE DATASETS

To show the superiority of CRPO, we compare its performance with two other static audio preference datasets: Audio-Alpaca (Majumder et al., 2024) and BATON (Liao et al., 2024) (see Appendix A.13 for details).

We apply preference optimization to TANGOFLUX-base, for one iteration since Audio-Alpaca and BATON are fixed datasets. Table 3 reports objective metrics $FD_{openl3}$, $KL_{passt}$, $CLAP_{score}$ and human evaluation results. Since all these models are variants of TANGOFLUX, we conducted human evaluation using 50 prompts from the AudioCaps test set.

Table 2: Human evaluation results on OVL and REL; `SA Open` := `Stable Audio Open`.

| Model | z-scores | | Ranking | | | | Elo | |
|---|---|---|---|---|---|---|---|---|
| | OVL | REL | OVL | | REL | | OVL | REL |
| | | | Mean | Mode | Mean | Mode | | |
| AudioLDM 2 | -0.3020 | -0.4936 | 3.5 | 4 | 3.7 | 4 | 1,236 | 1,196 |
| SA Open | 0.0723 | -0.3584 | 2.4 | 1, 3 | 3.3 | 3 | 1,444 | 1,268 |
| Tango 2 | -0.019 | 0.1602 | 2.4 | 2 | 1.9 | 2 | 1,419 | 1,507 |
| TANGOFLUX | **0.2486** | **0.6919** | **1.7** | **2** | **1.1** | **1** | **1,501** | **1,628** |

Table 3: Comparison of TANGOFLUX checkpoints aligned with different preference datasets. Along with objective metrics, we report human evaluation results – z-scores and Elo ratings. We embolden the **best** and underline the second-best scores for each metric.

| Model | Objective Metrics | | | z-scores | | Elo | |
|---|---|---|---|---|---|---|---|
| | $FD_{openl3}$ | $CLAP_{score}$ | $KL_{passt}$ | OVL | REL | OVL | REL |
| TANGOFLUX | **75.1** | **0.480** | **1.15** | **0.17** | **0.18** | **1,546** | **1,520** |
| TANGOFLUX-crpo-1 | 79.1 | 0.453 | 1.18 | 0.12 | 0.07 | 1,446 | 1,467 |
| TANGOFLUX-base | 80.2 | 0.431 | 1.22 | -0.06 | -0.21 | 1,325 | 1,253 |
| TANGOFLUX-alpaca | 80.0 | 0.448 | 1.20 | -0.02 | -0.00 | 1,428 | 1,366 |
| TANGOFLUX-baton | 80.5 | 0.437 | 1.20 | -0.21 | -0.04 | 1,253 | 1,392 |

Our results demonstrate that preference optimization with the CRPO dataset outperforms both Audio-Alpaca and BATON across all objective and subjective metrics. This highlights that CRPO is a highly effective approach for constructing audio preference datasets for optimization. TANGOFLUX also shows superior performance over TANGOFLUX-crpo1 in both subjective and objective benchmarks, highlighting the effectiveness of the iterative process in CRPO.

## 4.4 BATCHED ONLINE DATA GENERATION IS KEY

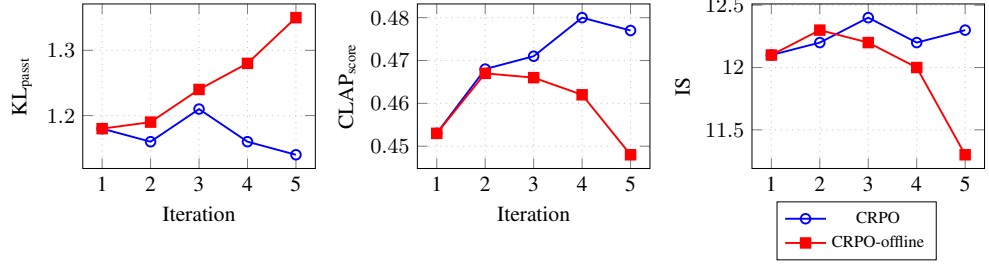

Figure 2: Trajectory of $CLAP_{score}$, IS and $KL_{passt}$ across training iterations: offline training peaks early at the peaking $CLAP_{score}$, increasing $KL_{passt}$ and decreasing IS ; in contrast, the $CLAP_{score}$ of online training keeps increasing until iteration 4, while $KL_{passt}$ has a clear downward trend. The IS value of online training also shows an gradual upward trend.

In Fig. 2, we present the results of five training iterations of CRPO, both with and without generating new data at each iteration. Our findings suggest that training on the same dataset over multiple iterations leads to quick performance saturation and eventual degradation. Specifically, for offline CRPO, the $CLAP_{score}$ decreases after the second iteration, while the $KL_{passt}$ increases significantly. The IS also peaked at iteration 2 of the offline CRPO and decreases after that. By the final iteration, the performance degradation is evident from the $CLAP_{score}$, $KL_{passt}$ and IS scores that are worse than in the first iteration, emphasizing the limitations of offline data. In contrast, the online CRPO with data generation before each iteration outperforms the offline CRPO w.r.t. $CLAP_{score}$, $KL_{passt}$ and IS.

This performance degradation could be ascribed to reward over-optimization (Rafailov et al., 2024a; Gao et al., 2022). Kim et al. (2024a) showed that the reference model serves as a regularizer in DPO training for language models. Several iterations of updating the reference model with the same dataset thus may hamper the due regularization of the loss. In Section 4.5, we show the paradoxical performance degradation with loss minimization, indicating over-optimization.

## 4.5 $\mathcal{L}_{\text{CRPO}}$ vs $\mathcal{L}_{\text{DPO-FM}}$

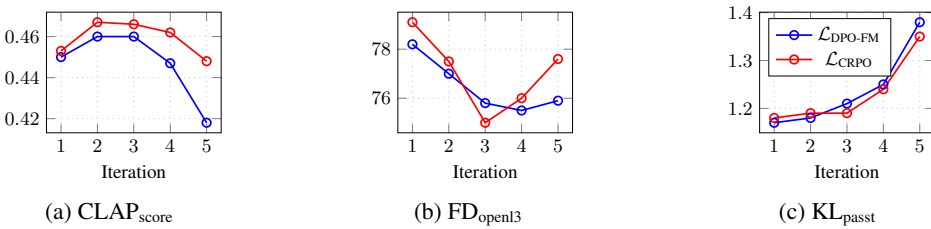

(a) CLAP$_{\text{score}}$      (b) FD$_{\text{openl3}}$      (c) KL$_{\text{passt}}$

Figure 3: Comparing $\mathcal{L}_{\text{DPO-FM}}$ and $\mathcal{L}_{\text{CRPO}}$ w.r.t. (a) CLAP$_{\text{score}}$, (b) FD$_{\text{openl3}}$, and (c) KL$_{\text{passt}}$ across iterations.

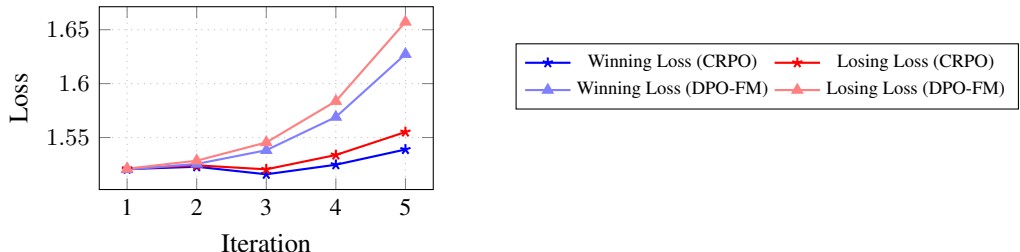

Figure 4: Winning and losing losses of $\mathcal{L}_{\text{DPO-FM}}$ and $\mathcal{L}_{\text{CRPO}}$ across iterations. Both losses increase with their margin.

To study the relationship between the winning and losing losses of $\mathcal{L}_{\text{CRPO}}$ and $\mathcal{L}_{\text{DPO-FM}}$ (see Eq. (2)), we calculate the average winning and losing losses of the final checkpoint (epoch 8) of each iteration on the training set. The losses are plotted in Fig. 4. Simultaneously in Fig. 3, we present the benchmark performances of the checkpoints by $\mathcal{L}_{\text{CRPO}}$ and $\mathcal{L}_{\text{DPO-FM}}$ on *AudioCaps* training set. Here, we only use fixed preference data by **TANGOFLUX**-base.

As shown in Fig. 4, the winning and losing losses of both $\mathcal{L}_{\text{CRPO}}$ and $\mathcal{L}_{\text{DPO-FM}}$ increase with each iteration, along with their difference/margin. Despite the increase in losses, Fig. 3 shows that benchmark performance improves, with $\mathcal{L}_{\text{CRPO}}$ achieving superior results in CLAP$_{\text{score}}$ while maintaining similar KL$_{\text{passt}}$ and FD$_{\text{openl3}}$ across all iterations. We observe a notable acceleration in loss growth from $\mathcal{L}_{\text{DPO-FM}}$ after iteration 3, which may indicate performance saturation or degradation. In contrast, $\mathcal{L}_{\text{CRPO}}$ exhibits a more gradual and stable increase in loss, maintaining a smaller margin and more controlled growth, leading to less performance degradation as compared to $\mathcal{L}_{\text{DPO-FM}}$. This highlights the role of the *winning loss* as a regularizer of the optimization dynamics by preventing the increase in margin at the cost of unmitigated increase of both *winning loss* and *losing loss*.

Our findings of increase in winning and losing losses in tandem with the margin is consistent with aligning LLMs with DPO (Rafailov et al., 2024b). This paradoxical performance improvement from both $\mathcal{L}_{\text{CRPO}}$ and $\mathcal{L}_{\text{DPO-FM}}$ is also noted by Rafailov et al. (2024a) in the context of LLMs.

## 4.6 INFERENCE TIME VS PERFORMANCE

**TANGOFLUX** beats the other models in terms of performance per unit of inference time, measured w.r.t. CLAP$_{\text{score}}$ and FD$_{\text{openl3}}$. For a fair comparison of inference time, we use a batch size of 1 for all our inference as some models do not support batch inference. See Appendix A.9 for more details.

## 5 RELATED WORKS

**Text-To-Audio Generation.** TTA Generation has lately drawn attention due to AudioLDM (Liu et al., 2024b; 2023), Tango (Majumder et al., 2024; Ghosal et al., 2023; Kong et al., 2024), and Stable Audio (Evans et al., 2024a;c;b) series of models. These adopt the diffusion framework (Song & Ermon, 2020; Rombach et al., 2022; Song et al., 2022; Ho et al., 2020), which trains a latent diffusion model conditioned on textual embedding. Another common framework for TTA generation is flow matching which was employed in models such as VoiceBox (Le et al., 2023), AudioBox (Vyas et al., 2023), FlashAudio (Liu et al., 2024c). ETTA (gil Lee et al., 2024),GenAU (Ali et al., 2024) highlights the benefits of scaling both data—using synthetic captions—and model size to enhance TTA generation performance. Whereas AudioTurbo(Zhao et al., 2025), IMPACT(Huang et al., 2025) also explored generating high quality audio with fast inference speed.

**Alignment Method.** Preference optimization is the standard approach for aligning LLMs, achieved either by training a reward model to capture human preferences (Ouyang et al., 2022) or by using the LLM itself as the reward model (Rafailov et al., 2024c). Recent advances improve this process through iterative alignment, leveraging human annotators to construct preference pairs or utilizing pre-trained reward models. (Kim et al., 2024a; Chen et al., 2024; Gulcehre et al., 2023; Yuan et al., 2024). Verifiable answers can enhance the construction of preference pairs. For diffusion and flow-based models, Diffusion-DPO shows that these models can be aligned similarly (Wallace et al., 2023). However, constructing preference pairs for TTA is challenging due to the absence of "gold" audio for given text prompts and the subjective nature of audio. BATON (Liao et al., 2024) relies on human annotations, which is not scalable.

## 6 CONCLUSION

We introduce TANGOFLUX, a flow-based text-to-audio model aligned using synthetic preference data generated online during training. Objective and human evaluations show that TANGOFLUX produces audio more representative of user prompts than existing diffusion-based models, achieving state-of-the-art performance with significantly fewer trained parameters. These advancements make TANGOFLUX a practical and scalable solution for widespread adoption.

## ETHICAL CONSIDERATIONS

There is a possibility of misuse of the audio generation model in fabricating harmful multimedia content.

## REPRODUCIBILITY STATEMENT

We shall publicly release the implementation of model training, inference, and evaluation upon acceptance. We also mention the hyperparameters in the appendix. An anonymized implementation is shared in the supplementary materials.

## ACKNOWLEDGEMENT

This research/project is supported by the National Research Foundation, Singapore under its AI Singapore Programme (AISG Award No: AISG3-GV2023-010). Further support comes from the National Research Foundation, Singapore, under its National Large Language Models Funding Initiative (AISG Award No: AISG-NMLP-2024-005), NTU SUG project #025628-00001: Post-training to Improve Embodied AI Agents.

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

# A APPENDIX

## A.1 THE USE OF LARGE LANGUAGE MODELS

We employed LLMs to assist with two key tasks: ensuring grammatical accuracy and improving the paper's narrative flow.

## A.2 AUDIO SAMPLE COUNT PER PROMPT VS TTA PERFORMANCE

In this ablation study on the initial iteration of TangoFlux-base, we compared generating $N$ audio samples per prompt for preference dataset creation (by pairing the worst and best outputs) where $N = 2, 5, 10$. All setups were trained for 8 epochs from TangoFlux-base, and we tracked the CLAP, FD, and KL metrics—using the maximum CLAP score per epoch for evaluation. The results in Table 4 show slightly better performance (higher CLAP score, lower FD) when using more samples. The decision on the optimal $N$ largely depends on the specific application and available resources. If computation cost is a limiting factor, opting for a lower $N$ might be a worthwhile compromise.

Table 4: Audio sample count vs TTA performance.

|  | 2 | 5 | 10 |
|---|---|---|---|
| CLAP | 0.443 | 0.453 | 0.455 |
| FD | 80.0 | 79.1 | 77 |
| KL | 1.24 | 1.18 | 1.21 |

## A.3 CLAP AS A REWARD MODEL

To validate CLAP as a proxy reward model for evaluating audio output, we further evaluate TANGOFLUX under a CLAP-driven Best-of-$N$ policy, where $N \in \{1, 5, 10, 15\}$. We use CLAP *630k-audioset-best.pt* checkpoint to rank the generated audios. The results in Table 5 suggest that increasing $N$ yield better CLAP$_{\text{score}}$ and KL$_{\text{passt}}$ while FD$_{\text{openl3}}$ remains about the same. This indicates that the CLAP can identify well-aligned audio outputs that better represent the textual descriptions, without compromising diversity or quality, as implied by the lower KL$_{\text{passt}}$ and similar FD$_{\text{openl3}}$.

Table 5: Best-of-$N$ FD, KL, and CLAP$_{\text{score}}$.

| Model | N | FD$_{\text{openl3}} \downarrow$ | KL$_{\text{passt}} \downarrow$ | CLAP$_{\text{score}} \uparrow$ |
|---|---|---|---|---|
| | 1 | 75.0 | 1.15 | 0.480 |
| | 5 | 74.3 | 1.14 | 0.494 |
| TANGOFLUX | 10 | 75.8 | 1.08 | 0.499 |
| | 15 | 75.1 | 1.11 | 0.502 |
| | 1 | 108.4 | 1.11 | 0.447 |
| | 5 | 108.8 | 1.05 | 0.467 |
| Tango 2 | 10 | 108.4 | 1.08 | 0.474 |
| | 15 | 108.7 | 1.06 | 0.473 |

## A.4 RECTIFIED FLOWS

Given a latent representation of an audio sample $x_1$, a noise sample $x_0 \sim \mathcal{N}(\mathbf{0}, \mathbf{I})$, time-step $t \in [0, 1]$, we can construct a training sample $x_t$ where the model learns to predict a velocity $v_t = \frac{dx_t}{dt}$ that guides $x_t$ to $x_1$. While there exist several methods of constructing transport path $x_t$, we used rectified flows (RFs) (Liu et al., 2022), in which the forward process are straight paths between target distribution and noise distribution, defined in Eq. (4). It is empirically shown that rectified flows are more sample efficient and degrade less than other formulations, while consuming fewer sampling steps (Esser et al., 2024). The model $u(\mathbf{x}_t, t; \theta)$ directly regresses the ground truth velocity $\mathbf{v}_t$ using

Table 6: TANGOFLUX with different classifier free guidance (CFG) values.

| Model | Steps | CFG | $\text{FD}_{\text{openl3}} \downarrow$ | $\text{KL}_{\text{passt}} \downarrow$ | $\text{CLAP}_{\text{score}} \uparrow$ |
|---|---|---|---|---|---|
| | 50 | 3.0 | 77.7 | **1.14** | 0.479 |
| | 50 | 3.5 | 76.1 | **1.14** | **0.481** |
| TANGOFLUX | 50 | 4.0 | 74.9 | 1.15 | 0.476 |
| | 50 | 4.5 | 75.1 | 1.15 | 0.480 |
| | 50 | 5.0 | **74.6** | 1.15 | 0.472 |

the flow matching loss in Eq. (5).

$$x_t = (1 - t)x_1 + t\tilde{x}_0, v_t = \frac{dx_t}{dt} = \tilde{x}_0 - x_1, \tag{4}$$

$$\mathcal{L}_{\text{FM}} = \mathbb{E}_{x_1, x_0, t} \left\| u(x_t, t; \theta) - v_t \right\|^2. \tag{5}$$

## A.5 MODEL TRAINING

We pretrained TANGOFLUX on Wavcaps (Mei et al., 2024) dataset for 80 epochs with the AdamW (Loshchilov & Hutter, 2019), $\beta_1 = 0.9, \beta_2 = 0.95$, a max learning rate of $5 \times 10^{-4}$. We used a linear learning rate scheduler for 2000 steps. We used five A40 GPUs with a batch size of 16 on each device, resulting in an overall batch size of 80. After pretraining, TANGOFLUX was finetuned on the *AudioCaps* training set for 65 additional epochs. Several works find that sampling timesteps $t$ from the middle of its range $[0, 1]$ leads to superior results (Hang et al., 2024; Kim et al., 2024b; Karras et al., 2022), thus, we sampled $t$ from a logit-normal distribution with a mean of 0 and variance of 1, following the approach in (Esser et al., 2024). We name this version as TANGOFLUX-base.

During the alignment phase, we used the same optimizer, but an overall batch size of 48, a maximum learning rate of $10^{-5}$, and a linear warmup of 100 steps. For each iteration of CRPO, we train for 8 epochs and select the last epoch checkpoint to perform batched online data generation. We performed 5 iterations of CRPO due to the manifestation of performance saturation.

## A.6 EFFECT OF CFG SCALE

We conduct an ablation of the effect of CFG scale for TANGOFLUX and show the result in Table 6. It reveals a trade-off: higher CFG values improve $\text{FD}_{\text{openl3}}$ score (lower $\text{FD}_{\text{openl3}}$) but slightly reduce semantic alignment (CLAP score), which peaks at CFG=3.5. The results emphasize CFG=3.5 as the optimal balance between fidelity and semantic relevance.

## A.7 HEUN VS EULER SOLVER

Unlike straightforward Euler solver, Heun solver uses second order look-ahead to better estimate trajectory. However, we get slightly worse results as shown in Table 7. We speculate that it could be due to overoptimization, analogous to greedy decoding for token generation.

Table 7: Comparison between Heun and Euler solver for integration over velocity.

| | Heun | Euler |
|---|---|---|
| NFE | 100 | 100 |
| CLAP | 0.474 | 0.480 |
| KL | 1.21 | 1.16 |
| FD | 75.3 | 75.1 |

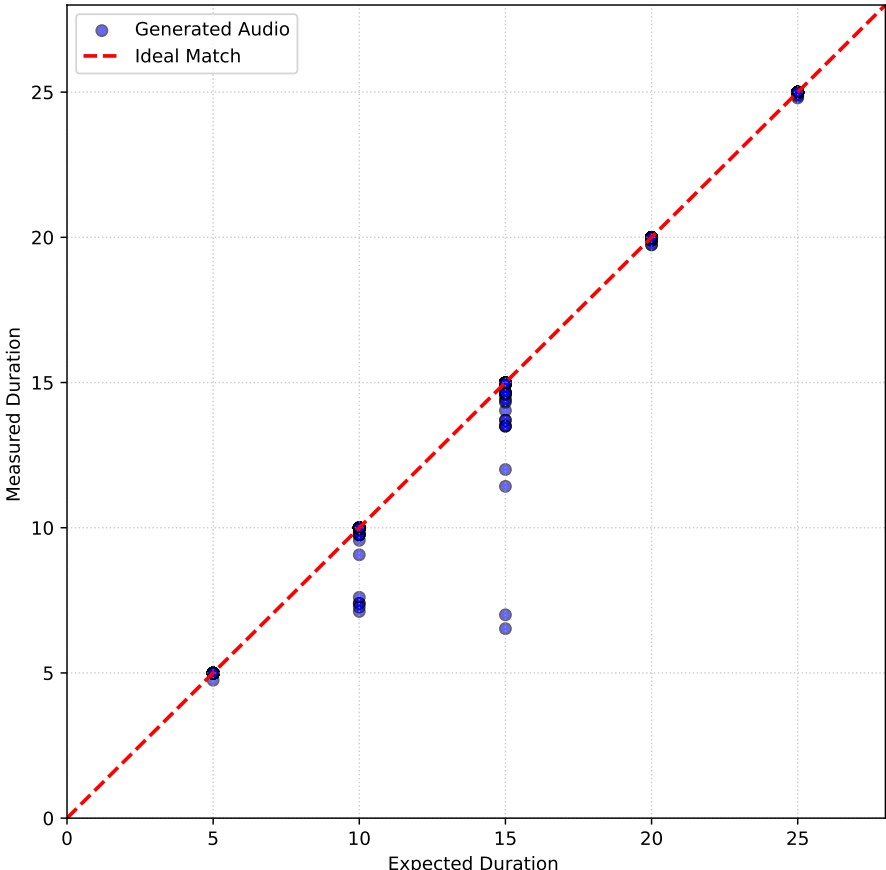

Figure 5: Expected input duration vs generated duration plot measured by an energy-based silence detection approach.

## A.8  ACCURACY OF DURATION CONTROL

We conducted an analysis to measure the accuracy of duration control and we adopted the approach in (Evans et al., 2024a) to measure the actual generated audio duration given the requested duration. We generated 100 audio samples each conditioned on textual prompts with requested durations of 5, 10, 15, 20, and 25 seconds. Since the generated audio is cropped at the requested duration, the actual duration of the content cannot exceed the requested duration. However, the model may introduce preceding silence within the generated sample. To accurately determine the actual duration of the non-silent, generated content, we employed an energy-based silence detection method on the audio waveforms. We visualize the scatter plot in Figure 5. Our plot suggests that most of the generated audios adhere exactly to the requested duration, with some minor exceptions.

## A.9  INFERENCE TIME VS PERFORMANCE COMPARISON

Across models, we compare the trajectory of $CLAP_{score}$ and $FD_{openl3}$ score with increasing inference time for steps 10, 25, 50, 100, 150, and 200, as shown in Figure 6. TANGOFLUX demonstrates a remarkable balance between efficiency and performance, consistently achieving higher $CLAP_{score}$ and lower $FD_{openl3}$ scores while requiring significantly less inference time compared to other models.

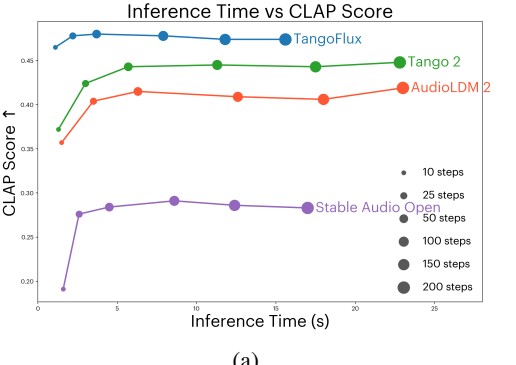 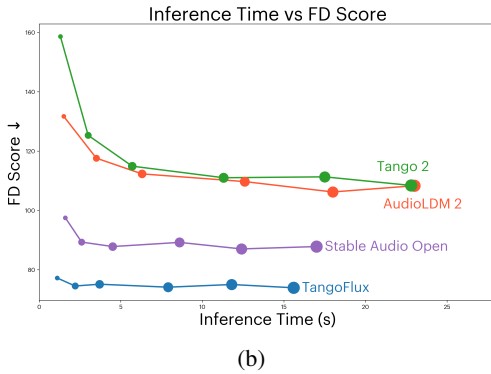

(a)                                                                                    (b)

Figure 6: Comparison of (a) CLAP$_{score}$ and (b) FD$_{openl3}$ vs Inference Time for each model. Results are plotted for step counts of 10, 25, 50, 100, 150, and 200.

For example, at 50 steps, **TANGOFLUX** achieves a CLAP$_{score}$ of 0.480 and an FD$_{openl3}$ score of 75.1 in just 3.7 seconds. In comparison, `Stable Audio Open` requires 4.5 seconds for the same step count but only achieves a CLAP$_{score}$ of 0.284 (41% lower than **TANGOFLUX**) and an FD$_{openl3}$ score of 87.8 (17% worse than **TANGOFLUX**). This demonstrates that **TANGOFLUX** achieves superior performance metrics in less time. Additionally, at a lower step count of 10, **TANGOFLUX** maintains strong performance with a CLAP$_{score}$ of 0.465 and an FD$_{openl3}$ score of 77.2 in just 1.1 seconds. In contrast, `Audioldm2` at the same step count achieves a lower CLAP$_{score}$ of 0.357 (23% lower) and a significantly worse FD$_{openl3}$ score of 131.7 (70% higher), while requiring 1.5 seconds (36% more time). We also observe that reducing the step count from 200 to 10 has a minimal impact on **TANGOFLUX**'s performance, highlighting its robustness. Specifically, **TANGOFLUX**'s CLAP$_{score}$ decreases by only 3.2% (from 0.480 to 0.465), and its FD$_{openl3}$ score increases by only 4.5% (from 73.9 to 77.2). In contrast, `Tango 2` shows a larger degradation, with its CLAP$_{score}$ decreasing by 16.0% (from 0.443 to 0.372) and its FD$_{openl3}$ score increasing by 37.8% (from 108.4 to 158.6).

These results highlight **TANGOFLUX**'s effectiveness in delivering high-quality outputs with lower computational requirements, making it a highly efficient choice for scenarios where inference time is critical.

## A.10    OBJECTIVE EVALUATION METRICS

KAD and FD$_{openl3}$ both evaluates the similarity between the statistics of a generated audio set and another reference audio set in the feature space. A low KAD, FD$_{openl3}$ indicates a realistic audio that closely resembles the reference audio. KL$_{passt}$ computes the KL divergence over the probabilities of the labels between the generated and the reference audio given the state-of-the-art audio tagger **PaSST**. A low KL$_{passt}$ signifies the generated and reference audio share similar semantics tags. CLAP$_{score}$ is a reference-free metric that measures the cosine similarity between the audio and the text prompt. High CLAP$_{score}$ score denotes the generated audio aligns with the textual prompt. IS measures the specificity and coverage of a set of samples. A high IS score represents the diversity in the generated audio. We use stable-audio-metrics (Evans et al., 2024a) to compute FD$_{openl3}$, KL$_{passt}$, CLAP$_{score}$ and AudioLDM evaluation toolkit (Liu et al., 2023) to compute **IS**. For KAD evaluation, we use the kadtk toolkit with the PANNs-WGLM (WaveGram-LogMel) embedding model, which has shown the highest correlation with human judgment. Note that we use different CLAP checkpoints to create our preference dataset (*630k-audioset-best*) and to perform evaluation (*630k-audioset-fusion-best*)[4]. These results are indicated in Table 1 as CLAP$_{score}$.

## A.11    CRPO VS GRPO

We have considered GRPO in our alignment objective. However, GRPO is a fully online algorithm which is computationally expensive. Hence, we conducted an experiment of optimizing GRPO loss

---

[4] https://huggingface.co/lukewys/laion_clap/blob/main/630k-audioset-fusion-best

on a static dataset that was constructed with TANGOFLUX-base (20k samples of 5 generations each, group size of 5). Our results show that the performance is slightly worse than running CRPO for 1 iteration.

Table 8: Comparison between CRPO and GRPO.

|  | CRPO | GRPO |
|---|---|---|
| CLAP | 0.453 | 0.448 |
| KL | 1.18 | 1.26 |
| FD | 79.1 | 79.6 |

## A.12   HUMAN EVALUATION

The human evaluation was performed using a web-based Gradio[5] app. Each annotator was presented with 20 prompts, each having four audio samples generated by four distinct text-to-audio models, shuffled randomly, as shown in Fig. 7. Before the annotation process, the annotators were instructed with the following directive:

---

Welcome *username*

# # Instructions for evaluating audio clips

**Please carefully read the instructions below**.

## ## Task

You are to evaluate four 10-second-long audio outputs to each of the 20 prompts below. These four outputs are from four different models. You are to judge each output with respect to two qualities:

- Overall Quality (OVL): The overall quality of the audio is to be judged on a scale from 0 to 100: 0 being absolute noise with no discernible feature. Whereas, 100 is perfect. **Overall fidelity, clarity, and noisiness of the audio are important here.**
- Relevance (REL): The extent of audio alignment with the prompt is to be judged on a scale from 0 to 100: with 0 being absolute irrelevance to the input description. Whereas, 100 is a perfect representation of the input description. **You are to judge if the concepts from the input prompt appear in the audio in the described temporal order.**

**You may want to compare the audios of the same prompt with each other during the evaluation.**

## ## Listening guide

1. Please use a head/earphone to listen to minimize exposure to the external noise.
2. Please move to a quiet place as well, if possible.

## ## UI guide

1. Each audio clip has two attributes OVL and REL below. You may select the appropriate option from the dropdown list.
2. To save your judgments, please click on any of the *save* buttons. All the *save* buttons function identically. They are placed everywhere to avoid the need to scroll to save.

Hope the instructions were clear. Please feel free to reach out to us for any queries.

---

[5]https://www.gradio.app

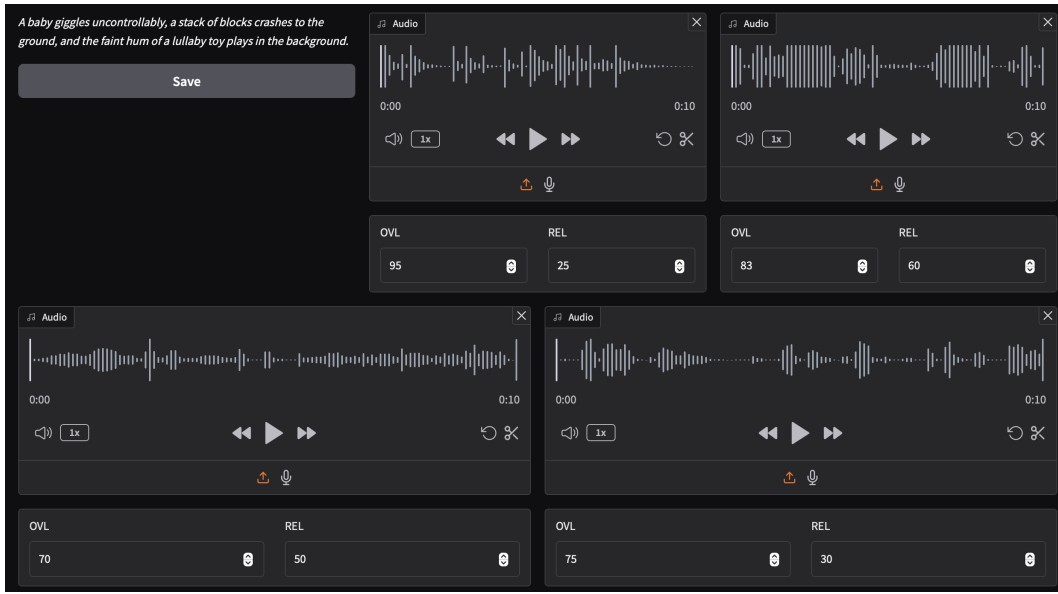

Figure 7: The Gradio-based human evaluation form created for the annotators to score the model generated audios with respect to the input prompts.

### A.12.1 EVALUATION DATASET

To evaluate the instruction-following capabilities and robustness of TTA models, we created 50 out-of-distribution complex captions, such as "*A pile of coins spills onto a wooden table with a metallic clatter, followed by the hushed murmur of a tavern crowd and the creak of a swinging door*". These captions describe 3–6 events and aim to go beyond conventional or overused sounds in the evaluation sets, such as simple animal noises, footsteps, or city ambiance. Events were identified using GPT4o to evaluate the captions generated. Each of the generated prompts contains multiple events including several where the temporal order of the events must be maintained. Details of our caption generation template and samples of generated captions can be found in the Appendix A.12.

### A.12.2 HUMAN EVALUATION METRICS

We report three key metrics for subjective evaluation:

$z$-**score:** The average of the scores assigned by individual annotators. Due to the subjective nature of these scores and the significant variance observed in the annotator scoring patterns, the ratings were normalized to z-scores at the annotator level: $z_{ij} = (s_{ij} - \mu_i)/\sigma_i$. $z_{ij}$: The z-score for annotator $i$'s score of model $M_j$. This is the score after applying z-score normalization. $s_{ij}$: The raw score assigned by annotator $i$ to model $j$. This is the original score before normalization. $\mu_i$: The mean score assigned by annotator $i$ across all models. It represents the central tendency of the annotator's scoring pattern. $\sigma_i$: The standard deviation of annotator $i$'s scores across all models. This measures the variability or spread in the annotator's ratings.

This normalization procedure adjusts the raw scores, centering them around the annotator's mean score and scaling by the annotator's score spread (standard deviation). This ensures that scores from different annotators are comparable, helping to mitigate individual scoring biases.

**Ranking:** Despite z-score normalization, the variability in annotator scoring can still introduce noise into the evaluation process. To address this, models are also ranked based on their absolute scores. We utilize the mean (average rank of a model), and mode (the most common rank of a model) as metrics for evaluating these rankings.

**Elo:** Elo-based evaluation, a widely adopted method in language model assessment, involves pairwise model comparisons. We first normalized the absolute scores of the models using z-score normalization and then derived Elo scores from these pairwise comparisons. Elo score mitigates the noise and

inconsistencies observed in scoring and ranking techniques. Specifically, Elo considers the relative performance between models rather than relying solely on absolute or averaged scores, providing a more robust measure of model quality under subjective evaluation. While ranking-based evaluation provides an ordinal comparison of models, determining the order of performance (e.g., Model A ranks first, Model B ranks second), it does not capture the magnitude of differences between ranks. For instance, if the difference between the first and second rankers is minimal, this is not evident from ranks alone. Elo scoring addresses this limitation by integrating both ranking and pairwise performance data. In ranking-based systems, the rank $R_i$ of a model $M_i$ is determined purely by its position relative to others:

$$R_i = \text{position of } M_i \text{ in the sorted list of models based on performance.}$$

However, this approach fails to quantify: 1) The gap in performance between consecutive ranks. 2) The consistency of relative performance across different pairwise comparisons. Elo scoring provides a probabilistic measure of model performance based on pairwise comparisons. By leveraging annotator scores, Elo assigns a continuous score $E_i$ to each model $M_i$, capturing its relative strength.

### A.12.3 PROMPTS USED IN THE EVALUATION

Table 9: Prompts used in human evaluation and their characteristics.

| Prompts | Multiple Events | Temporal Events |
|---|---|---|
| A robotic arm whirs frantically while an electric plasma arc crackles and a metallic voice counts down ominously, interspersed with glass vials clinking to the floor. | ✓ | ✓ |
| Unfamiliar chirps overlap with a low, throbbing hum as bioluminescent plants audibly crackle and squelch with movement. | ✓ | ✗ |
| Dripping water echoes sharply, a distant growl reverberates through the cavern, and soft scraping metal suggests something lurking unseen. | ✓ | ✗ |
| Alarms blare with rising urgency as fragments clatter against a metallic hull, interrupted by a faint hiss of escaping air. | ✓ | ✓ |
| Hundreds of tiny wings buzz with a chaotic pitch shift, joined by the faint clattering of mandibles and an organic squish as they collide. | ✓ | ✗ |
| Jagged rocks crumble underfoot while distant ocean waves crash below, punctuated by the sudden snap of a rope. | ✓ | ✓ |
| Digital beeps and chirps meld with overlapping chatter in multiple languages, as automated drones whiz past, scanning barcodes audibly. | ✓ | ✗ |
| Rusted swings creak in rhythmic disarray, a faint mechanical laugh stutters from a distant speaker, and the sound of gravel crunches under unseen footsteps. | ✓ | ✗ |
| Bubbling lava gurgles ominously, instruments beep irregularly, and faint crackling signals static from a failing radio. | ✓ | ✓ |
| Tiny pops and hisses of chemical reactions intermingle with the rhythmic pumping of a centrifuge and the soft whirr of air filtration. | ✓ | ✗ |

| | | |
|---|:---:|:---:|
| The faint hiss of a gas leak grows louder as metal chains rattle and a single marble rolls across the floor. | ✓ | ✓ |
| A hand slaps a table sharply, followed by the shuffle of playing cards and the hum of an overhead fan. | ✓ | ✓ |
| A train horn blares in the distance as a bicycle bell chimes and a soda can pops open with a fizzy hiss. | ✓ | ✗ |
| A drawer creaks open, papers rustle wildly, and the sharp click of a lock snapping shut echoes. | ✓ | ✗ |
| A burst of static interrupts soft typing sounds, followed by the distant chirp of a pager and a cough. | ✓ | ✓ |
| A heavy book thuds onto a desk, accompanied by the faint buzz of a fluorescent light and a muffled sneeze. | ✓ | ✗ |
| The sharp squeak of sneakers on a gym floor blends with the rhythmic bounce of a basketball and the screech of a metal door. | ✓ | ✗ |
| An elevator dings, its doors sliding open, as muffled voices overlap with the shuffle of heavy bags. | ✓ | ✗ |
| A clock ticks steadily, a light switch clicks on, and the crackle of a fire igniting briefly fills the silence. | ✓ | ✓ |
| A fork scrapes a plate, water drips slowly into a sink, and the faint hum of a refrigerator lingers in the background. | ✓ | ✗ |
| A cat hisses sharply as glass shatters nearby, followed by hurried footsteps and the slam of a closing door. | ✓ | ✓ |
| A parade marches through a town square, with drumbeats pounding, children clapping, and a horse neighing amidst the commotion. | ✓ | ✓ |
| A basketball bounces rhythmically on a court, shoes squeak against the floor, and a referee's whistle cuts through the air. | ✓ | ✗ |
| A baby giggles uncontrollably, a stack of blocks crashes to the ground, and the faint hum of a lullaby toy plays in the background. | ✓ | ✗ |
| The rumble of a subway train grows louder, followed by the screech of brakes and muffled announcements over a crackling speaker. | ✓ | ✓ |
| A beekeeper moves carefully as bees buzz intensely, a smoker puffs softly, and wooden frames creak as they're lifted. | ✓ | ✗ |
| A dog shakes off water with a noisy splatter, a bicycle bell rings, and a distant lawnmower hums faintly in the background. | ✓ | ✗ |
| Books fall off a shelf with a heavy thud, a chair scrapes loudly across a wooden floor, and a surprised gasp echoes. | ✓ | ✗ |
| A soccer ball hits a goalpost with a metallic clang, followed by cheers, clapping, and the distant hum of a commentator's voice. | ✓ | ✓ |

| | | |
|---|---|---|
| A hiker's pole taps against rocks, a mountain goat bleats sharply, and loose gravel tumbles noisily down a steep slope. | ✓ | ✓ |
| A rooster crows loudly at dawn, joined by the rustle of feathers and the crunch of chicken feed scattered on the ground. | ✓ | ✗ |
| A carpenter saws through wood with steady strokes, a hammer strikes nails rhythmically, and a measuring tape snaps back with a metallic zing. | ✓ | ✗ |
| A frog splashes into a pond as dragonflies buzz nearby, accompanied by the distant croak of toads echoing through the marsh. | ✓ | ✗ |
| The crack of a whip startles a herd of cattle, their hooves clatter against a dirt path as a rancher shouts commands. | ✓ | ✗ |
| A paper shredder whirs noisily, the rustle of documents being fed in grows louder, and a stapler clicks shut in rapid succession. | ✓ | ✗ |
| An elephant trumpets in the savanna as a herd stomps through dry grass, accompanied by the buzz of flies and the distant roar of a lion. | ✓ | ✗ |
| A mime claps silently as a juggling act clinks glass balls together, and a crowd bursts into laughter at the clatter of a dropped prop. | ✓ | ✗ |
| A train conductor blows a sharp whistle, metal wheels screech on the rails, and passengers murmur while settling into their seats. | ✓ | ✓ |
| A squirrel chitters nervously as acorns drop from a tree, landing with dull thuds, while leaves rustle above in quick bursts of movement. | ✓ | ✗ |
| A blacksmith hammers molten iron with rhythmic clangs, a bellows pumps air with a whoosh, and sparks sizzle on a stone floor. | ✓ | ✗ |
| A skateboard grinds loudly against a metal rail, followed by the sharp slap of wheels hitting pavement and a triumphant cheer from the rider. | ✓ | ✗ |
| An old typewriter clacks rapidly as paper rustles with each keystroke, interrupted by the sharp ding of the carriage return. | ✓ | ✗ |
| A pack of wolves howls in unison as dry leaves crunch underfoot, and the faint trickle of a nearby stream echoes through the forest. | ✓ | ✗ |

## A.13 BATON AS A PREFERENCE DATASET

BATON contains human-annotated data where annotators assign a binary label of 0 or 1 to each audio sample based on its alignment with a given prompt: 1 indicates alignment, while 0 indicates misalignment. We construct a preference dataset by pairing audio samples labeled 1 (winners) with those labeled 0 (losers) for same prompt, creating a set of winner-loser pairs.

