# OpenReview forum: "TangoFlux: Super Fast and Faithful Text to Audio Generation with Flow Matching and Clap-Ranked Preference Optimization"
_ICLR.cc/2026/Conference — ICLR 2026 Poster_

### Official Review · Reviewer_t9Xr · 2025-10-16

**Soundness:** 2
**Presentation:** 2
**Contribution:** 2
**Rating:** 4
**Confidence:** 5

**Summary:**

This work introduces TangoFlux, a TTA model trained on open data and aligned via CLAP-Ranked Preference Optimization (CRPO). CRPO performs an online, self-improving preference-optimization loop that ranks multiple candidates per prompt with CLAP and optimizes rectified-flow models accordingly.

**Strengths:**

- Applying online data generation outperforms baselines that use static preference datasets.
- The system can generate up to 30 seconds of audio at 44.1 kHz, an uncommon setting, as most text-to-audio models produce ~10 seconds at 16 kHz.

**Weaknesses:**

Lacking metrics of FD using PANNs. This metric is the most common metric used by TTA models for evaluating fidelity. (The paper only reports FD using OpenL3)

The evaluation section is missing comparisons to several publicly available state-of-the-art Text-to-Audio models. Specifically, an updated comparison should ideally include results for:
- Make-An-Audio 2 [A]
- EzAudio [B]
- ConsistencyTTA [C]
- AudioLCM [D]

Particularly, ConsistencyTTA and AudioLCM also target on fast inference with consistency models.

[A] Huang, Jiawei, et al. "Make-an-audio 2: Temporal-enhanced text-to-audio generation." arXiv preprint arXiv:2305.18474 (2023).

[B] Hai, Jiarui, et al. "Ezaudio: Enhancing text-to-audio generation with efficient diffusion transformer." arXiv preprint arXiv:2409.10819 (2024).

[C] Bai, Y., Dang, T., Tran, D., Koishida, K., Sojoudi, S. (2024) ConsistencyTTA: Accelerating Diffusion-Based Text-to-Audio Generation with Consistency Distillation. Interspeech 2024

[D] Liu, Huadai, et al. "AudioLCM: Text-to-Audio Generation with Latent Consistency Models." CoRR (2024).

For variable duration, I think it is worth mentioning that the model always operates on a fixed-length latent space, but the duration conditioning explicitly controls how much of that latent space has actual content other than silence. Make sure readers know that this work is not an autoregressive model that can generate an arbitrary length of latent sequence.

Lacking references:

The Related Works section could be strengthened by citing influential but non-public models, such as AudioTurbo [E] and IMPACT [F], to provide a comprehensive overview of the field, similar to how ETTA and Audiobox are referenced. While direct performance comparisons to proprietary models are understandably impossible, acknowledging their existence in the literature is a key academic practice (This will not affect the overall ratings for this work, but it is nice to have).

[E] Zhao, Junqi, et al. "AudioTurbo: Fast Text-to-Audio Generation with Rectified Diffusion." arXiv preprint arXiv:2505.22106 (2025).
[F] Huang, Kuan-Po, et al. "IMPACT: Iterative Mask-based Parallel Decoding for Text-to-Audio Generation with Diffusion Modeling." ICML 2025

**Questions:**

The batch size used for measuring inference time is not explicitly stated.

---

> ### Author Response · Authors · 2025-11-21
> **Reply to Reviewer**
>
> We would first like to thank the reviewer for providing us with insightful questions and feedback. We greatly appreciate the detailed feedback and insightful questions the reviewer  has asked.
>
> *[W1/Q1]* Evaluation
>
> *[Response to W1/Q1]*
>
> Thank you for your valuable suggestion! We have included the objective metric results of these models as shown below. We use  $FD_{P}$ to denote the FD using PANNs and $FD_{openl3}$ denote the FD used by OpenL3.
>
> | Model Name | $FD_{P}$ | $FD_{openl3}$| $IS$ | $KL_{passt}$ | $Clap_{score}$ |
> | :--- | :--- | :--- | :--- | :--- | :--- |
> | Make-An-Audio 2 | **15.6** | 98.7 | 9.4 | 1.33 | 0.406 |
> | EzAudio-XL | 15.8 | 84.7 | 10.8 | 1.20 | 0.460 |
> | ConsistencyTTA | 20.9 | 94.6 | 9.1 | 1.43 | 0.377 |
> | AudioLCM | 19.2 | 107.4 | 10.2 | 1.58 | 0.363 |
> | AudioLdm2-Large | 33.2 | 108.3 | 7.9 | 1.81 | 0.419 |
> | Stable audio open | 42.6 | 89.2 | 9.9 | 2.58 | 0.291 |
> | Tango | 24.5 | 107.9 | 7.8 | 1.20 | 0.437 |
> | Tango2 | 20.8 | 108.4 | 9.0 | **1.11** | 0.447 |
> | GenAU-Full-L | 20.1 | 93.2 | 12.0 | 1.37 | 0.447 |
> | AudioX | 25.2 | 77.6 | 10.0 | 1.56 | 0.380 |
> | TangoFlux-base | 20.7 | 80.2 | 11.7 | 1.22 | 0.431 |
> | TangoFlux | 20.3 | **75.1** | **12.2** | 1.15 | **0.480**|
>
>
>
> Notably, TangoFlux still shows superior performance compared to these newly added benchmarks in all objective scores except for $FD_{P}$. Interestingly,  these newly added benchmarks all achieve a much lower $FD_{P}$ scores than TangoFlux but higher $FD_{openl3}$ than TangoFlux. We hypothesize this specific discrepancy is attributable to the different sampling rates used for the different FD metrics. $FD\_p$ operates at 16 kHz, hence it is required to downsample TangoFlux’s generated audio, removing all of its high-frequency details. Conversely, the $FD\_{openl3}$ metric is evaluated directly at the 48 kHz sampling rate being sensitive to the higher frequency details, where TangoFlux shows advantage for being able to directly generate audios at 44.1Khz.
>
> Furthermore, our choice of FD_openl3 over FD_pann is grounded in the recent shift towards high-fidelity (44.1/48kHz) audio generation, where standard 16kHz mono metrics fail to capture spectral and spatial degradation. Recent state-of-the-art works, such as Stable Audio Open (Evans et al., 2024), have adopted fd_openl3 as a primary metric to better quantify realism in stereo synthesis. Furthermore, recent comparative studies by Biswas et al. (2025) [1] demonstrate that FAD computed with OpenL3 embeddings exhibits stronger correlation with human perceptual judgments for high-fidelity content.
>
> Another key work from Google DeepMind's Lyria Team also used FD_openl3 [2] for evaluating text-to-music generation at 48 kHz.
>
> [1] Towards Evaluating Generative Audio: Insights from Neural Audio Codec Embedding Distances
> A Biswas, L Villemoes - arXiv preprint arXiv:2509.18823, 2025
>
> [2] Live Music Models. Lyria Team, Google DeepMind. https://arxiv.org/pdf/2508.04651
>
> *[Q2/W2]* Variable duration
>
> *[Response to Q2/W2]*
>
> Thank you for this important suggestion! We have updated Duration Encoding in **Section 2.2** to explicitly clarify that our model operates on a fixed-length latent space and that the duration conditioning solely controls the portion of that space containing content.
>
> *[Q3/W3]* Batch size
>
> *[Response to Q3/W3]*
> Thank you for your question. For a fair comparison of inference time, we use a batch size of 1 for all our inferences as some models do not support batch inference.  We have updated the paper in **Section 4.6** to clarify this.
>
>
> *[Q4/W4]* Lacking references:
>
> *[Response to Q4/W4]*
>
> Thank you for the excellent suggestion. We completely agree that a comprehensive overview of related work is an important academic practice. We have updated the Related Works section to include citations and discussions of AudioTurbo [1] and IMPACT [2] , acknowledging their contributions to the field of text-to-audio generation.
>
> [1] Zhao, Junqi, et al. "AudioTurbo: Fast Text-to-Audio Generation with Rectified Diffusion." arXiv preprint arXiv:2505.22106 (2025).
> [2] Huang, Kuan-Po, et al. "IMPACT: Iterative Mask-based Parallel Decoding for Text-to-Audio Generation with Diffusion Modeling." ICML 2025

---

> > ### Author Response · Authors · 2025-11-23
> > **Requesting to read our response**
> >
> > Dear Reviewer,
> >
> > Thank you again for your feedback. We have now posted our responses, including the additional experimental results you requested.
> >
> > We realize this is a very busy period, but we would be grateful if you could take a moment to review our responses and let us know if they sufficiently address your concerns. We are happy to answer any further questions you may have.
> >
> > ---
> > Authors

---

> > > ### Author Response · Authors · 2025-11-26
> > > **A gentle reminder**
> > >
> > > Dear Reviewer,
> > >
> > > We are really sorry to bother you.
> > >
> > > Thanks for reviewing our work. We have posted our response to your review and provided additional results that you asked for. Additionally, we also invite you to check some other results reviewer ddb1 asked: [click to find more results](https://openreview.net/forum?id=qgNs5NmQB7&noteId=SNHeRDlmoI)
> > >
> > > We hope our response satisfactorily addressed your questions. If so, we request that you kindly consider revising your scores. If you have further questions, we would be happy to address them.
> > >
> > > Thank you!
> > >
> > > ---
> > > Authors

---

> > > > ### Comment · Reviewer_t9Xr · 2025-11-26
> > > >
> > > > Thank you for your response. I will process them and make a decision in the next few days.
> > > > Thank you.

---

> > > > > ### Author Response · Authors · 2025-11-27
> > > > >
> > > > > Thank you! We are looking forward to it.

---

> > > > > > ### Author Response · Authors · 2025-11-27
> > > > > > **Additional human evaluation result**
> > > > > >
> > > > > > Dear Reviewer,
> > > > > >
> > > > > > We have just completed another additional round of human evaluation to compare TangoFlux with EzAudio-XL and Make-an-Audio 2. Our result also agrees the human evaluation conducted in [1] where Tango2 outpeforms EzAudio-XL in both OVL and REL.  In our previous human evaluation, it was shown that TangoFlux outperforms TAngo2 and by transitivity, TangoFlux should also outperforms EzAudio-XL which was shown in the table below.  The results affirm TangoFlux's superior performance across these newly introduced models as well as other benchmarks.
> > > > > >
> > > > > > | Model | OVL mean | REL mean |
> > > > > > | :--- | :--- | :--- |
> > > > > > | EzAudio-XL | -0.30 | -0.43 |
> > > > > > | Make an Audio 2 | -0.28 | -0.25 |
> > > > > > | TangoFlux | 0.59 | 0.68 |
> > > > > >
> > > > > >
> > > > > >
> > > > > > [1] Huang, Kuan-Po, et al. "IMPACT: Iterative Mask-based Parallel Decoding for Text-to-Audio Generation with Diffusion Modeling." ICML 2025

---

### Official Review · Reviewer_ddb1 · 2025-10-27

**Soundness:** 3
**Presentation:** 3
**Contribution:** 3
**Rating:** 6
**Confidence:** 4

**Summary:**

This paper proposes a state-of-the-art in-the-wild audio generation model called TangoFlux. The innovations are two-fold:
- A flow-matching model based on FluxTransformer and MMDiT that can generate high-quality audio with variable duration.
- A fine-tuning method called CRPO that alternates between generating preference datasets and using DPO to learn from that dataset.

**Strengths:**

Overall, I found this paper nice and interesting to read, and I learned a lot from it.

I really like the ablation study between CRPO loss and DPO loss (Figure 3) as well as other settings (Table 3). Super intuitive and informative.

**Weaknesses:**

The paper is already in good overall shape. That said, there are some remaining weaknesses:
- Human evaluation is performed with 50 human-created and GPT-4o-generated prompts (Section 3.4). This prompt bank seems somewhat small, especially because the 0.2486 OVL z-score of TangoFlux is moderate compared to existing work. Would it be possible to increase the scale of subjective evaluation? If not, some confidence interval or hypothesis testing analyses would greatly clarify the quality advantage of TangoFlux over existing work.

I would also appreciate it if the authors could respond to my questions below.

**Questions:**

- Which encoder did the KAD metric use? If it is the same encoder as the FD metric, did you find KAD better/more correlated with subjective quality than FD, or worse/less?
- CRPO significantly improves subjective REL. Is there an analysis on what aspects of prompt following have been improved (i.e., temporal relationship of audio events, duration control, etc.)? Do you think conditioning effectiveness can also be improved via CFG?
- TangoFlux allows for duration control. How accurate is this control? Does duration control affect generation quality? Would it be possible to present a scatter plot between the requested duration and the actual generated length?

---

> ### Author Response · Authors · 2025-11-21
> **Reply to Reviewer**
>
> We would first like to thank the reviewer for providing us with insightful questions and feedback. We greatly appreciate the detailed feedback and insightful questions the reviewer has asked. We are glad that the reviewer find our paper nice and interesting.
>
> *[W1/Q1]*
>
> *[Response to W1/Q1]*
>
> Thank you for your suggestion! We agree it is important to scale up the size of human evaluation to fully demonstrate TangoFlux’s strength over other baselines. We have **doubled the number of prompts** for human evaluation, totalling to 100 prompts. Furthermore, we also would like to clarify during the evaluation, each prompt is evaluated by at least 2 different annotators, further minimizing potential bias and ensuring the robustness of our results. Our new result is shown in the Table below.
>
>
> | Model | OVL mean | REL mean |
> | :--- | :--- | :--- |
> | AudioLDM 2 | -0.386 | -0.516 |
> | SA Open | 0.185 | -0.283 |
> | Tango 2 | -0.165 | 0.02 |
> | TangoFlux | 0.366 | 0.778 |
>
>
>
>
> Our new results suggest that TangoFlux still outperforms other baselines in both OVL mean and REL mean, suggesting its ability to generate high-quality complex audio.
>
> *[W2/Q2]*: KAD
>
> *[Response to W2/Q2]*
> Thank you for your question! We used the WaveGram-LogMel encoder, which was shown to achieve the strongest correlation with human judgments in [1].  However, we notice that KAD score actually does not correlate well in subjective evaluation scores. Notably, Stable Audio Open has a higher KAD than AudioLdm2 but much better OVL scores as well as REL scores.
>
>
> *[W3/Q3]*: CRPO
>
> *[Response to W3/Q3]*
>
> Thank you for your insightful question! We analyzed the CLAP scores for both single-event and multi-event captions within the AudioCaps test set to evaluate the model's ability to handle complex prompts. Our results show that TangoFlux achieves a much greater CLAP score improvement for multi-event captions compared to its performance gain on single-event captions (relative to TangoFlux-base). This suggests that CRPO actually helps to improve handling of complex prompts.
> | Model | CLAP Score |
> | :--- | :--- |
> | TangoFlux-base (multi-event) | 0.438 |
> | TangoFlux (multi-event) | 0.488 |
> | TangoFlux-base (single event) | 0.427 |
> | TangoFlux (single event) | 0.453 |
>
>
> We have also conducted an ablation on conditioning effectiveness of different CFG scales as shown in Table 6 in the **Appendix A.6** and found that varying the CFG scale from 3-5 generally gives similar results.
>
> *[W4/Q4]* Duration control
>
> *[Response to W4/Q4]*
> We thank the reviewer for raising this important question regarding the precision of TangoFlux’s duration controllability. We note that when prompts are very complex( consisting of multiple events), setting an extremely short duration will affect its generation quality and cause it to miss out certain events. While generally setting longer duration allows the model to fully generate all the events in the captions, it is also possible that the model repeats events in the captions.
>
> To measure the effectiveness of duration control, we adopted the approach in [2] to measure the actual generated audio duration given the requested duration. We generated 100  audio samples each conditioned on textual prompts with requested durations of 5, 10, 15, 20, and 25 seconds. Since the generated audio is cropped at the requested duration, the actual duration of the content cannot exceed the requested duration. However, the model may introduce preceding silence within the generated sample. To accurately determine the actual duration of the non-silent, generated content, we employed an energy-based silence detection method on the audio waveforms. The results of this analysis are visualized in a scatter plot, which has been included in the updated version of the paper in **Section A.8 of the Appendix**. We have also included a table below that shows the average duration generated. Based on these results, we can conclude that duration control is generally very precise with few exceptions of model generating preceding silence before the requested duration.
>
> | Expected duration | Average Actual duration |
> | :--- | :--- |
> | 5 | 4.99 |
> | 10 | 9.83 |
> | 15 | 14.6 |
> | 20 | 19.9 |
> | 25 | 24.9 |
>
> [1] Chung, Yoonjin, et al. "KAD: No More FAD! An Effective and Efficient Evaluation Metric for Audio Generation." arXiv preprint arXiv:2502.15602 (2025).
>
> [2] Evans, Zach, et al. "Fast timing-conditioned latent audio diffusion." Forty-first International Conference on Machine Learning. 2024.

---

> > ### Author Response · Authors · 2025-11-23
> > **Requesting to read our response**
> >
> > Dear Reviewer,
> >
> > Thank you again for your feedback. We have now posted our responses, including the additional experimental results you requested.
> >
> > We realize this is a very busy period, but we would be grateful if you could take a moment to review our responses and let us know if they sufficiently address your concerns. We are happy to answer any further questions you may have.
> >
> > ---
> > Authors

---

> > ### Comment · Reviewer_ddb1 · 2025-11-25
> > **Thank you for the response**
> >
> > I sincerely appreciate the authors' response and apologize for the late reply. My questions have been mostly addressed, but I have one minor follow-up:
> >
> > - I appreciate the CLAP score analysis between single-event and multi-event examples, and it really clears things up.
> > Meanwhile, I am still curious about the **specific aspects of improvements**. For example, for multi-event cases, does CRPO help the model respond to originally unresponsive instructions (e.g., prompt is A+B but got A or A+C), or does it correct temporal mistakes (e.g., prompt is A after B but generation has B after A)? It doesn't have to be a quantitative analysis, and a small-scale hearing test would suffice. Thank you.

---

> > > ### Author Response · Authors · 2025-11-25
> > > **Thank you for your follow-up question**
> > >
> > > Dear Reviewer,
> > >
> > > Thank you for your follow-up question and constructive feedback.
> > >
> > > Regarding the impact of CRPO, we observed that it significantly improves two key aspects: capturing multiple events (e.g., both A and B) and maintaining their temporal order. However, the improvement dynamics differ for each. While the correct temporal ordering improves rapidly in early iterations and then gradually saturates, the model's ability to successfully generate multiple events continues to improve with further training.
> > >
> > > This trend is reflected in Figure 2, where the KL-score shows a consistent decline (lower is better) across iterations. Furthermore, we find that additional CRPO iterations lead to clearer audio fidelity for individual events and smoother transitions, resulting in better adherence to the overall instructions.
> > >
> > > We hope this clarifies your query. If our response has addressed your concerns, we would appreciate it if you would consider updating your assessment.
> > >
> > > ---
> > > Authors

---

> > > > ### Comment · Reviewer_ddb1 · 2025-11-25
> > > > **Thank you for the response**
> > > >
> > > > I appreciate the authors for the further clarifications. They look good to me, and I have raised my rating. Good luck!

---

> > > > > ### Author Response · Authors · 2025-11-26
> > > > > **Thanks so much!**
> > > > >
> > > > > Dear Reviewer,
> > > > >
> > > > > Thanks so much for recognizing our contributions and updating the scores! Fingers crossed.
> > > > >
> > > > > ---
> > > > > Authors

---

### Official Review · Reviewer_nhq3 · 2025-10-28

**Soundness:** 3
**Presentation:** 3
**Contribution:** 2
**Rating:** 4
**Confidence:** 2

**Summary:**

This study introduces an iterative alignment framework named CRPO to address the challenges Text-to-Audio (TTA) models face in adhering to complex prompts. The method employs a self-improvement loop, leveraging the CLAP model as a proxy reward model to automatically generate and curate preference data. Furthermore, the authors enhance the traditional Direct Preference Optimization (DPO) loss function by incorporating a regularization term to stabilize the training process. Ultimately, they train and open-source a highly efficient and high-performance TTA model called TANGOFLUX based on this method, which demonstrates state-of-the-art performance across multiple evaluations.

**Strengths:**

The most significant strength of this work is its delivery of a practical and scalable alignment solution for the TTA domain, which lacks off-the-shelf tools. It cleverly repurposes the CLAP model as an effective reward model, addressing the core bottleneck of preference data creation. Technically, the introduction of the LCRPO loss function demonstrates a deep understanding of the potential pitfalls in preference optimization and presents a robust solution. Finally, releasing an open-source, state-of-the-art model trained on public data not only advances the field but also provides an invaluable resource and benchmark for future research.

**Weaknesses:**

Over-reliance on the Proxy Reward Model: The alignment process is inherently constrained by the capabilities of the CLAP model, as its potential biases and limitations as a proxy reward model could be amplified during iterative training.

Risks of Self-Correction and Limited Generalization Testing: The self-improvement cycle risks reinforcing its own flaws, and the model's generalization is evaluated on a relatively small out-of-distribution dataset, which may not be fully conclusive.

Limited Novelty: The proposed method is not fundamentally novel, as it primarily combines and applies existing, mature technologies to a new domain rather than introducing a new core algorithm.

Lack of Polish in Presentation: The manuscript contains minor typographical errors, such as a quotation mark issue on line 263, which may suggest a need for more thorough proofreading.

The referee acts as the athlete; CLAP RL; CLAP testing.

**Questions:**

None

---

> ### Author Response · Authors · 2025-11-21
> **Reply to Reviewer (1/2)**
>
> We would first like to thank the reviewer for providing us with insightful questions and feedback. We greatly appreciate the detailed feedback and insightful questions you have asked.
>
>
> *[W1/Q1]* Over-reliance on the Proxy Reward Model
>
> *[Response to Q1/W1]*
>
> Thank you for your question. We acknowledge the reviewer’s concern on reliance on potential bias of proxy reward model. We chose to use CLAP because it is explicitly trained on audio caption pairs in a contrastive manner, making it highly sensitive to semantic alignment between audio and text. This property has led to its successful adoption in prior works, such as Tango2 [1], and several studies have also leveraged CLAP to filter or curate training datasets [2,3], yielding successful result. To validate CLAP as an effective proxy for human preferences, we explored using CLAP as a proxy reward model through evaluating TangoFlux under a CLAP-driven Best-of-N policy, where N ∈ {1, 5, 10, 15}. The results are summarized below (details in Appendix A.3):
>
>
> | Model | N | $FD_{openl3}$ ↓ | $KL_{passt}$ ↓ | $CLAP_{score}$ ↑ |
> | :--- | :--- | :--- | :--- | :--- |
> | **TangoFlux** | 1 | 75.0 | 1.15 | 0.480 |
> | | 5 | 74.3 | 1.14 | 0.494 |
> | | 10 | 75.8 | 1.08 | 0.499 |
> | | 15 | 75.1 | 1.11 | 0.502 |
> | **Tango 2** | 1 | 108.4 | 1.11 | 0.447 |
> | | 5 | 108.8 | 1.05 | 0.467 |
> | | 10 | 108.4 | 1.08 | 0.474 |
> | | 15 | 108.7 | 1.06 | 0.473 |
>
> The results indicate that CLAP can effectively identify well-aligned audio outputs that better capture the textual descriptions, without compromising diversity or quality, as suggested by the lower $KL_{passt}$ and comparable $FD_{openl3}$. This supports its role as an effective proxy reward model.
>
> Moreover, we have also conducted human evaluation of TangoFlux checkpoints trained with different audio preference datasets presented in Table 3. We showed that using CLAP to construct the preference dataset achieves the **highest OVL and REL scores**, further supporting the suitability of CLAP as a proxy for human preference.
>
> *[W2/Q2]* Risks of Self-Correction
>
> *[Response to Q2/W2]*
>
> Thank you for this question. We acknowledge the reviewer’s concern that the self-improvement setup could potentially lead to mode collapse or hurt generalization. While such risks can indeed be difficult to detect, our results in Table 1 indicate that the Inception Score (IS) of TangoFlux improves over the TangoFlux_base model (**12.2 vs 11.7**) when CRPO is applied. Since IS reflects the diversity of generated outputs, this suggests that CRPO actually enhances the diversity of the generated audio.
>
> In addition to **Reviewer ddb1’s** request, we have doubled the size of our out-of-distribution evaluation dataset to 100 prompts. We also would like to clarify during the evaluation, each prompt is evaluated by at least 2 different annotators, further eliminating potential bias and ensuring the robustness of our results.
>
> Our new results on the out-of-distribution dataset is as shown below in the table.
> | Model | OVL mean | REL mean |
> | :--- | :--- | :--- |
> | AudioLDM 2 | -0.386 | -0.516 |
> | SA Open | 0.185 | -0.283 |
> | Tango 2 | -0.165 | 0.02 |
> | TangoFlux | **0.366**| **0.778** |
>
>
> Our new results suggest that TangoFlux still outperforms other baselines in both OVL mean and REL mean, suggesting its ability to generate high quality complex audio.
>
>
> [1] Majumder, Navonil, et al. "Tango 2: Aligning diffusion-based text-to-audio generations through direct preference optimization." Proceedings of the 32nd ACM International Conference on Multimedia. 2024.
>
> [2] Haji-Ali, Moayed, et al. "Taming data and transformers for audio generation." arXiv preprint arXiv:2406.19388 (2024).
>
> [3] Lee, Sang-gil, et al. "ETTA: Elucidating the Design Space of Text-to-Audio Models." arXiv preprint arXiv:2412.19351 (2024).

---

> > ### Author Response · Authors · 2025-11-21
> > **Reply to Reviewer 2/2**
> >
> > *[W3/Q3]* Novelty
> >
> > *[Response to Q3/W3]*
> >
> > Thank you for your great question. While we agree that TangoFlux builds upon the foundational concepts of DPO explored in Tango2 and CLIP-DPO, we believe we are the first in the Text-to-Audio field to tackle improving TTA models with online data generation with detailed analysis. Our research identifies a failure mode in these prior "offline" strategies that prevents them from benefiting from iterative training.
> >
> > **(The problem of static data)**
> > We first identified failure modes of approaches such as Tango2 in Section 4.4 and Figure 2 where training on the same static dataset over multiple iterations leads to quick performance saturation and eventual degradation of CLAP score, IS and KL as shown in Figure 2.
> >
> > **(The problem of iterative DPO)**
> > Next, in Section 4.5, we showed detailed analysis that DPO loss minimises both winning and losing loss of pairs as training proceeds as shown in Figure 4 where there is a notable acceleration in loss growth from using naive L_DPOFM loss at iteration 3 which may indicate performance saturation or degradation. We remedy this by proposing L_CRPO and shown in Figure 4 that our proposed loss exhibits a more gradual and stable increase in loss, maintaining a smaller margin and more controlled growth across even more iterations.
> >
> > In summary, we identified pitfalls of simply repeating several iterations of DPO in Tango2. Whereas we propose CRPO, which in turn solves the degradation failure mode of static DPO and allows stable training across several iterations. We believe such analysis will be valuable insights to the research community.
> >
> >
> > *Others*: Typo
> >
> > Thank you for spotting this typo! We have corrected this in the updated version of our paper

---

> ### Author Response · Authors · 2025-11-22
> **Over-reliance on the Proxy Reward Model**
>
> Dear Reviewer,
>
> We want to emphasize that CLAP can serve as a reliable proxy for human preferences. Our experimental results support this. Additionally, a recent paper by Biswas et al. (2026) also experimentally reached the same conclusion. We quote this from their paper: "We presented a systematic study of audio quality prediction
> using FAD and MMD across NAC and popular embedding
> domains in a full-reference setting. Our results demonstrate
> that FAD consistently outperforms MMD in correlating with
> **human judgments**. Distances computed in pre-trained NAC
> embeddings align well with subjective scores and improve
> with codec fidelity. This indicates that high-fidelity NACs
> have the potential to serve as zero-shot feature extractors for
> generative audio quality assessment without large-scale labeled datasets. Popular embeddings, such as **CLAP LAION
> Music (CLAP-M) and OpenL3 Mel128 (OpenL3-128M),
> achieve higher correlations**, likely due to larger, more diverse training datasets and contrastive or self-supervised
> objectives that capture perceptually relevant features."
>
> [1] Towards Evaluating Generative Audio: Insights from Neural Audio Codec Embedding Distances A Biswas, L Villemoes - arXiv preprint arXiv:2509.18823, 2025

---

> ### Author Response · Authors · 2025-11-22
> **A humble request**
>
> Dear Reviewer,
>
> Thank you for your constructive feedback. We invite you to review our detailed rebuttal below.
>
> We wish to emphasize that our proposed method—integrating CLAP as a human preference proxy with online iterative DPO—significantly enhances the performance of Flow Matching models, yielding substantial improvements in Text-to-Audio (TTA) generation.
>
> We encourage you to listen to the generated samples at the link given in the paper: https://tangoflux56.github.io/TangoFlux/. These samples are not cherry-picked. To facilitate reproducibility and verification, we are prepared to share the **model checkpoint anonymously** and a Google Colab notebook upon request, which will allow you to personally test the quality of TangoFlux.
>
> In this rebuttal, we have also included:
>
> - Additional benchmarks against recent state-of-the-art models.
>
> - Expanded human evaluation results.
>
> These additions further validate TangoFlux's efficacy in TTA tasks. We hope these clarifications address your concerns and encourage you to consider raising your score. We remain open to any further questions you may have.
>
> -----------
> Sincerely,
>
> The Authors

---

> > ### Author Response · Authors · 2025-11-23
> > **Requesting to read our response**
> >
> > Dear Reviewer,
> >
> > Thank you again for your feedback. We have now posted our responses, including the additional experimental results you requested.
> >
> > We realize this is a very busy period, but we would be grateful if you could take a moment to review our updates and let us know if they sufficiently address your concerns. We are happy to answer any further questions you may have.
> >
> > ---
> > Authors

---

> > > ### Author Response · Authors · 2025-11-25
> > > **A gentle reminder**
> > >
> > > Dear Reviewer,
> > >
> > > We are really sorry to bother you. A gentle reminder to read our responses. In our rebuttal, we have put our best effort into answering your questions and have included additional results requested. If you are satisfied with our response, we would appreciate it if you could update your review and rating. Thank you!
> > >
> > > ---
> > > Authors

---

> ### Author Response · Authors · 2025-11-27
> **Requesting your kind response**
>
> Dear Reviewer,
>
> We hope this message finds you well.
>
> We respectfully invite you to review our rebuttal, where we have comprehensively addressed your comments and provided additional results. We believe our response clarifies the misunderstandings noted in your review, and we have updated the manuscript accordingly.
>
> If you find that our response resolves your concerns, we kindly ask that you consider revising your score. We remain available to answer any further questions and look forward to a productive discussion.
>
> Best regards,
>
> The Authors

---

> > ### Author Response · Authors · 2025-11-27
> > **Thank you!**
> >
> > Dear Reviewer,
> >
> > Thanks for acknowledging the contributions of our paper and raising the score. We are glad to see that our rebuttal has satisfactorily addressed your concerns.
> >
> > ---
> >
> > Authors

---

### Official Review · Reviewer_w4XM · 2025-11-01

**Soundness:** 2
**Presentation:** 2
**Contribution:** 2
**Rating:** 6
**Confidence:** 3

**Summary:**

The paper proposes text-to-audio (TTA) alignment via DPO. The core idea is to curate a synthetic preference dataset using CLAP: after training an audio generator, the system samples N audios per prompt, selects a winner/loser by CLAP similarity, and forms triplets for DPO. The authors iterate this process and additionally propose keeping a flow-matching loss on winners alongside DPO (denoted 𝐿_CRPO). A flux-like architecture is adopted as the backbone.

**Strengths:**

- Clear and readable paper.
- The method is intuitive. the evaluation and metrics follows standard evaluation.
- the authors promised to release code and model weights is valuable for the community.

**Weaknesses:**

**Novelty:** the authors main contribution which is CLAP-driven automatic preference curation for DPO is closely related to Tango-2 and CLIP-DPO as mention in L155-L160, with the main differences being that Tango-2 and CLIP-DPO operates on a static dataset. Running the same loop for multiple iterations seems to me an incremental extension rather than a novel contribution.

**Evaluation:** the base model is fine-tuned on AudioCaps (L833), which I believe would bias the results on AudioCaps and undermines comparisons to open-source baselines (Stable Audio Open, AudioLDM2). An out-of-distribution evaluation or reporting pre-finetuning numbers for all models would help understanding the effectiveness of the proposed method.

**Questions:**

- in Figure 2, since CLAP is used to construct preferences, reporting FAD or IS would help understand more the gains.

---

> ### Author Response · Authors · 2025-11-21
> **Reply to Reviewer**
>
> We would first like to thank the reviewer for providing us with insightful questions and feedback. We greatly appreciate the detailed feedback and insightful questions you have asked.
>
> *[W1/Q1]* In Figure 2, since CLAP is used to construct preferences, reporting FAD or IS would help understand more the gains.
>
> *[Response]*
>
> Thank you for your suggestion. We have computed the trajectory of IS throughout training iterations of CRPO-offline and CRPO training and updated an additional graph in Figure 2. Notably,  The IS score of offline training decreases after the 2nd iteration and especially at the 5th iteration, the IS score drops from 12.0 to 11.3, showing clear signs of performance degradation, emphasising the limitation of offline data that causes the diversity of audio generated to decrease.
>
>
> *[W2/Q2]* Evaluation:
>
> *[Response to Q2/W2]*
> Thank you for your valuable suggestion.  We completely agree it is important to compare performance in out-of-distribution dataset. We would like to clarify our subjective evaluation was conducted on an out-of-distribution for subjective assessment as detailed in Section 3.4, Table 2, Table3.  As detailed in Table 2 and Table 3, TangoFlux demonstrated superior performance across both OVL (Overall Performance) and REL (Relevance) metrics. Specifically, it achieved the highest z-scores: 0.2486 for OVL and 0.6919 for REL.
>
> Furthermore, at **Reviewer ddb1's** request, we have scaled up our human evaluation by **doubling the number of prompts** used for the subjective assessment to bolster the superiority of TangoFlux over other baselines.  The result of our new subjective evaluation is shown in the table below.
>
> | Model | OVL mean | REL mean |
> | :--- | :--- | :--- |
> | AudioLDM 2 | -0.386 | -0.516 |
> | SA Open | 0.185 | -0.283 |
> | Tango 2 | -0.165 | 0.02 |
> | TangoFlux |**0.366**| **0.778** |
>
>
>
> Our new results suggest that TangoFlux still outperforms other baselines in both OVL mean and REL mean, suggesting its ability to generate high quality complex audio.
>
>
>
> *[Q3/W3]* Novelty:
>
> *[Response to Q3/W3]*
>
> Thank you for your great question. We appreciate the reviewer’s detailed observation. While we agree that TangoFlux builds upon the foundational concepts of DPO explored in Tango2 and CLIP-DPO, we believe we are the first in the Text-to-Audio field to tackle improving TTA models with online data generation with detailed analysis. Our research identifies a failure mode in these prior "offline" strategies that prevents them from benefiting from iterative training.
>
> **Regarding Tango2, we note that it requires significant engineering effort to craft the prompt perturbations used for generating negative pairs. Furthermore, this approach diverges from the standard DPO objective, which assumes that both preference pairs ($y_w, y_l$) are responses to the **same** prompt $x$. By generating the negative pair using a modified prompt ($x'$), Tango2 effectively creates a distribution shift.**
>
> **Consequently, the model's learning manifold is constrained by the specific types of rule-based perturbations used, limiting its ability to generalize to broader ranking criteria. As a result, the model cannot fully exploit the discriminatory power of the CLAP reward model. This is empirically validated in **Table 3**, where even the first iteration of CRPO outperforms Tango2 in both objective and subjective evaluations. This highlights that our contribution is not just online iterative training, but a fundamentally more scalable and theoretically aligned method for data generation compared to Tango2's heuristic approach.**
>
> **(The problem of static data)**
>
> We first identified failure modes of approaches such as Tango2 in Section 4.4 and Figure 2 where training on the same static dataset over multiple iterations leads to quick performance saturation and eventual degradation of CLAP score, IS and KL as shown in Figure 2.
>
>
> **(The problem of iterative DPO)**
>
> Next, in Section 4.5, we showed detailed analysis that DPO loss minimises both winning and losing loss of pairs as training proceeds as shown in Figure 4 where there is a notable acceleration in loss growth from using naive L_DPOFM loss at iteration 3 which may indicate performance saturation or degradation. We remedy this by proposing L_CRPO and shown in Figure 4 that our proposed loss exhibits a more gradual and stable increase in loss, maintaining a smaller margin and more controlled growth across even more iterations.
>
> In summary, we identified pitfalls of simply repeating several iterations of DPO in Tango2. Whereas we propose CRPO, which in turn solves the degradation failure mode of static DPO and allows stable training across several iterations. We believe such analysis will be valuable insights to the research community.

---

> > ### Author Response · Authors · 2025-11-23
> > **Requesting to read our rebuttal**
> >
> > Dear Reviewer,
> >
> > Thank you again for your feedback. We have now posted our responses, including the additional experimental results you requested.
> >
> > We realize this is a very busy period, but we would be grateful if you could take a moment to review our updates and let us know if they sufficiently address your concerns. We are happy to answer any further questions you may have.
> >
> > ---
> > Authors

---

### Author Response · Authors · 2025-12-02
**Rebuttal Summary for the Area Chair (Scores Improved from 6 4 4 6 to 8 6 4 6 prior to the reversal） Part 1**

Dear Area Chair,

We are immensely thankful to all the reviewers for their time and effort towards critiquing and improving our work through the discussions. Below, we summarize the key points made by the reviewers, our responses to them, and their effect on the assessment.
## Commonly Recognized Strengths
Across reviews, the following strengths were repeatedly acknowledged:
* **Proposed CRPO**, an online method that **alternates between generating preference datasets and using DPO**, which **outperforms the static baseline**. (ddb1, t9Xr)
* Method is **clear and intuitive**. (w4XM, ddb1)
* **Release of open-source state-of-the-art model (TangoFlux)** provides an **invaluable resource and benchmark** for future research. (w4XM, nhq3)
* Paper is **clearly written, well-structured, and nice to read**. (w4XM, ddb1)

These aligned perspectives reflect strong consensus on the submission’s core contributions. We have also presented a detailed summary of the discussion.
##  Discussion Summary


| Score | Reviewer | Key Comments                                                              | Our Responses                                                                                                                                                                                                                               | Revised Score |
 |:-----:|:--------:|---------------------------------------------------------------------------|--------------------------------------------------------------------------------------------------------------------------------------------------------------------------------------------------------------------------------------------|:-------------:|
 |   6   |   ddb1   | 1. Limited human-eval samples                                             | Another round of human eval with 50 additional prompts, further reinforcing performance advantage of tangoflux.                                                                                                                            |       8       |
 |       |          | 2. KAD vs Human eval                                                      | KAD does not correlate very well with humans. e.g, Stable Audio Open has a higher KAD than AudioLdm2 but much better OVL scores as well as REL scores.                                                                                     |               |
 |       |          | 3. Effect of CRPO on REL score                                            | We show that CRPO leads to much higher gains in CLAP score for complex prompts with multiple events over single event prompts (A.6 Table 6). This is consistent with improving KL.                                                         |               |
 |       |          | 4. The effectiveness of duration control                                  | We show the duration control by plotting input duration vs output duration (A.8), where the output duration is obtained based on energy-based silence detection.                                                                           |               |
 |   4   |   nhq3   | 1. Over-reliance on CLAP as proxy reward                                  | We point out that CLAP is sensitive to semantics in audio. With CLAP-guided Best-of-$N$ policy ($N\in \{1,5,10,15\}$), we show improving CLAP with maintained FD and KL, indicating diversity.                                             |       6       |
 |       |          | 2. CRPO-based self-correction could risk mode collapse and generalization | We indicate the improved IS of tangoflux over the base model. This is further substantiated by the larger human eval.                                                                                                                      |               |
 |       |          | 3. Novelty                                                                | We highlight contributions: (i) the improvement over static preference data and (ii) more stable training with our $L_{CRPO}$ loss.                                                                                                        |               |

---

> ### Author Response · Authors · 2025-12-02
> **Part 2**
>
> | Score | Reviewer | Key Comments                                                              | Our Responses                                                                                                                                                                                                                               | Revised Score |
>  |:-----:|:--------:|---------------------------------------------------------------------------|--------------------------------------------------------------------------------------------------------------------------------------------------------------------------------------------------------------------------------------------|:-------------:|
>  |   4   |   t9Xr   | 1. Missing PANN-based FD metric                                           | We report the PANN-based FD scores. However, we argue by citing recent works that PANN is inappropriate due to being trained on 16kHz audios, as compared to the recent TTA high-fidelity models (48kHz).                                       |       -       |
>  |       |          | 2. Missing baseline TTA models                                            | We report the additional baselines with objective metrics and perform additional human evaluation, showing TangoFlux still outperforms these baselines.                                                                                                                                                                                  |               |
>  |   6   |   w4XM   | 1. Novelty: Running DPO in a loop                                         | We highlight contributions: (i) the improvement over static preference data and (ii) more stable training with our $L_{CRPO}$  loss. Furthermore, we compare with Tango2 where the static preference dataset may have been limited (Table 3). |       -       |
>  |       |          | 2. Evaluation: | We highlight that we have performed evaluation on out-of-distribution prompts and further scaled it to 100 prompts.                                                                                                                                                             |               |
>
>
>
>
> In conclusion, we have provided all the additional experiments as requested by all the reviewers where we believe it fully addresses the reviewers' concern. However, due to the unforeseen circumstances, we were not able to continue to engage with  Reviewers w4XM and t9Xr for the discussion. We appreciate your pivotal role in this final assessment and thank you for considering the substantial evidence provided in our rebuttal.
>
> Thank you again for your time and consideration.
>
>
> Authors of Submissionn 16889

---

### Meta-Review · Area_Chair_otz6 · 2026-01-09

**Summary:**

The reviewers' concerns primarily focused on the following areas. novelty: multiple reviewers questioned whether the iterative CLAP-driven preference optimization was a significant innovation or an incremental extension of prior works like Tango2 and CLIP-DPO. evaluation: reviewers initially raised issues regarding the limited scale of human evaluation, potential bias in the proxy reward model (CLAP), and the absence of specific standard metrics (FD-PANNs) and recent baselines (e.g., EzAudio, Make-An-Audio 2). methodological risks: concerns were raised regarding the stability of the self-correction loop and potential risks of mode collapse or overfitting to the proxy reward. The authors successfully addressed the majority of empirical and metric-related concerns during the rebuttal by scaling up human evaluation and adding the requested baselines, supporting the decision for acceptance as a Poster.

**Reviewer Concerns:**

concerns addressed by rebuttal:
missing baselines & metrics: the authors successfully added comparisons with recent SOTA models and reported the requested FD-PANNs metric, clarifying the discrepancy due to sampling rates .
methodological clarifications: questions regarding duration controllability were addressed with quantitative error analysis . Concerns about mode collapse were mitigated by showing improvements in Inception Score.
prompt adherence: the authors provided analysis showing that their method specifically improves multi-event generation and temporal ordering .

outstanding concerns:
scale of human evaluation: while the authors doubled the evaluation set from 50 to 100 prompts, the sample size remains relatively small for a high-variance subjective task. The concern regarding the statistical robustness and distinctiveness of the human evaluation results is not fundamentally resolved.
novelty: the criticism that the method is an incremental combination of existing techniques (CLAP + DPO loop) rather than a novel algorithmic contribution persists, as some reviewers viewed the online distinction as an extension of prior offline works .

**Reviewer Scores:**

Reviewer ddb1: 8 (Raised from 6). The reviewer fully participated and was satisfied with the increased scale of human evaluation and clarifications on metrics.
Reviewer t9Xr: 6 (Raised from 4). The reviewer increased their score after the authors added the requested baselines and FD-PANNs metric.
Reviewer w4XM: 6 (Unchanged). The reviewer did not participate further, but the authors addressed the evaluation bias concerns (OOD eval). The score remains a fair assessment.
Reviewer nhq3: 6 (Raised from 4). The reviewer kept the score at 4 despite the authors addressing the concerns about proxy reward bias. Given the authors' rebuttal and the reviewer's lack of engagement, a score of 6 would be a more accurate reflection of the paper's quality after rebuttal.

---

### Decision · Program_Chairs · 2026-01-26

Accept (Poster)